# A Differential and Pointwise Control Approach to Reinforcement Learning

**Minh Nguyen**
University of Texas at Austin
minhpnguyen@utexas.edu

**Chandrajit Bajaj**
University of Texas at Austin
bajaj@cs.utexas.edu

## Abstract

Reinforcement learning (RL) in continuous state-action spaces remains challenging in scientific computing due to poor sample efficiency and lack of pathwise physical consistency. We introduce Differential Reinforcement Learning (Differential RL), a novel framework that reformulates RL from a continuous-time control perspective via a differential dual formulation. This induces a Hamiltonian structure that embeds physics priors and ensures consistent trajectories without requiring explicit constraints. To implement Differential RL, we develop Differential Policy Optimization (dfPO), a pointwise, stage-wise algorithm that refines local movement operators along the trajectory for improved sample efficiency and dynamic alignment. We establish pointwise convergence guarantees, a property not available in standard RL, and derive a competitive theoretical regret bound of $\mathcal{O}(K^{5/6})$. Empirically, dfPO outperforms standard RL baselines on representative scientific computing tasks, including surface modeling, grid control, and molecular dynamics, under low-data and physics-constrained conditions.

## 1 Introduction

Reinforcement learning (RL) has achieved notable success across domains such as robotics, biological sciences, and control systems ([16, 20, 6, 2]). Yet, its application to scientific computing remains limited largely due to persistent challenges in sample complexity, lack of physical consistency, and weak theoretical guarantees. Unlike supervised learning, where models learn from labeled datasets, RL agents must explore through trial-and-error, often receiving sparse and delayed feedback. This makes data efficiency a critical bottleneck. Furthermore, standard RL methods typically fail to encode physical laws or structural priors, leading to suboptimal solutions in scientific problems governed by continuous dynamics.

Model-based RL (MBRL) [32] improves sample efficiency by learning a surrogate model of the environment. However, current approaches require information typically unavailable under scientific computing settings. For example:

1. **Explicit reward model:** Many MBRL algorithms (e.g., SVG [14], iLQR [29], PILCO [10]) require access to the whole reward functions and/or its derivative, when in reality reward functionals are only available at points along the explored trajectories.

2. **Re-planning assumptions:** Shooting methods and other trajectory-based planners [23] often assume the ability to re-plan from a particular intermediate time step. However, in many scientific computing problems, agents must always generate trajectories starting from the initial time, without resetting or modifying the trajectory midway (see Section C.7 for an example).

As a result, researchers often revert to model-free RL combined with customized reward shaping. However, such approaches still fail to incorporate physics-informed priors or recover optimal policies

under limited sample budgets. This motivates a fundamentally different approach: Rather than optimizing cumulative rewards over discrete steps, we interpret RL through the lens of continuous-time control, viewing trajectory returns as integrals and introducing a differential dual formulation. This naturally gives rise to a Hamiltonian structure that embeds inductive biases aligned with physical dynamics, even when explicit physical constraints are absent.

To implement the framework, we develop Differential Policy Optimization (dfPO), an algorithm that directly optimizes a local trajectory operator and refines policy behavior pointwise along the trajectory, rather than through global value estimates. This locality enables dfPO to align updates with system dynamics at each timestep, avoiding inefficient exploration that is both costly and misaligned with physical constraints. By maintaining consistency with the optimal trajectory throughout execution, dfPO minimizes redundant relearning and reduces sample waste. This is crucial in scientific settings where simulation cost and rollout horizon are tightly constrained.

We evaluate Differential RL on a suite of scientific problems that involve complex dynamics, implicit objectives, and limited data, the settings where traditional RL struggles.

1. **Surface Modeling:** Optimization over evolving surfaces, where rewards depend on the geometric and physical properties of the surface.

2. **Grid-based Modeling:** Control on coarse grids with fine-grid evaluations, representative of multiscale problems with implicit rewards.

3. **Molecular Dynamics:** Learning in graph-based atomic systems where dynamics depend on nonlocal interactions and energy-based cost functionals.

**Contributions.** Our main contributions are:

1. We introduce Differential RL, a reinforcement learning framework that replaces cumulative reward maximization with trajectory operator learning, naturally embedding physics-informed priors via a differential dual formulation.

2. We propose Differential Policy Optimization (dfPO), a policy optimization algorithm with rigorous pointwise convergence guarantees and a regret bound of $\mathcal{O}(K^{5/6})$, enabling effective learning in low-data, physics-constrained environments.

3. We validate dfPO across diverse scientific tasks, demonstrating strong performance over several standard RL baselines.

**Organization.** In Section 2, we introduce the new framework of differential reinforcement learning and the associated algorithm called dfPO. In Section 3, we outline the theoretical pointwise convergence theorem and regret analysis for the dfPO algorithm with explicit details given in the Appendix. In Section 4, we apply the dfPO algorithm to three representative scientific-computing tasks, and show competitive performance against popular RL benchmarks.

## 2 Differential reinforcement learning

### 2.1 Problem formulation

In standard reinforcement learning, an agent operates in a Markov Decision Process (MDP) defined by the 5-tuple $(\mathcal{S}, \mathcal{A}, \mathbb{P}, r, \rho_0)$, where $\mathcal{S}$ and $\mathcal{A}$ are the set of states and actions respectively, $\mathbb{P} : \mathcal{S} \times \mathcal{A} \times \mathcal{S} \to \mathbb{R}$ is the transition probability distribution, $r : \mathcal{S} \times \mathcal{A} \to \mathbb{R}$ is the reward function, $\rho_0 : \mathcal{S} \to \mathbb{R}$ is the distribution of initial state. At step $k$, the agent choose an action $a_k \in \mathcal{A}$ given its current state $s_k \in \mathcal{S}$ based on $\pi(a_k|s_k)$:

$$s_0 \sim \rho_0(s_0), a_k \sim \pi(a_k|s_k), s_{k+1} \sim \mathbb{P}(s_{k+1}|s_k, a_k) \tag{1}$$

The goal is to maximize the expected cumulative reward $\mathcal{J} = \mathbb{E}_\pi \left[ \sum_{k=0}^{H-1} r(s_k, a_k) \right]$. For a given policy $\pi$, the associated value function is defined as:

$$V_\pi(s) = \mathbb{E}_{a, s_1, \dots} \left[ \sum_{k=0}^{H-1} r(s_k, a_k) | s_0 = s \right] \tag{2}$$

The (optimal) value function is then defined as $V(s) := \arg\max_\pi V_\pi(s)$. Many reinforcement learning algorithms revolve around estimating and improving this value function. However, instead of remaining in this discrete-time formulation, we shift to a continuous-time viewpoint. By associating each discrete step $k$ with a timestamp $t_k = k\Delta_t$ and setting the terminal time $T = t_H = H\Delta_t$, we approximate the discrete sum with a time integral:

$$\max_\pi \mathbb{E}\left[\sum_{k=0}^{H-1} r(s_k, a_k)\right] = \max_\pi \mathbb{E}\left[\sum_{k=0}^{H-1} r(s_{t_k}, a_{t_k})\right] \approx \max_\pi \mathbb{E}\left[\int_0^T r(s_t, a_t)dt\right] \qquad (3)$$

This leads to the control-theoretic objective:

$$\max_\pi \mathbb{E}\left[\int_0^T r(s_t, a_t)dt\right] \text{ subject to } \dot{s}_t = f(s_t, a_t) \qquad (4)$$

where $f$ denotes the transition dynamics. Instead of directly solving this constrained optimization, we invoke Pontryagin's Maximum Principle [17], which introduces a dual formulation analogous to the Hamiltonian framework in classical mechanics. We augment the system with an adjoint variable $p$, and define the **Hamiltonian function** $\mathcal{H}$ through the Legendre transform:

$$\mathcal{H}(s, p, a) := p^T f(s, a) - r(s, a) \qquad (5)$$

Let $a^*(s, p)$ denotes the optimal action as a function of state and adjoint variables. Substituting this back gives the **reduced Hamiltonian function** $\hbar$:

$$\hbar(s, p) := \mathcal{H}(s, p, a^*(s, p)) \qquad (6)$$

The resulting **differential dual system** imposes the following constraints on the trajectory:

$$\begin{bmatrix} \dot{s} \\ \dot{p} \end{bmatrix} = \begin{bmatrix} \frac{\partial \hbar}{\partial p}(s, p) \\ -\frac{\partial \hbar}{\partial s}(s, p) \end{bmatrix} \text{ subject to } \hbar(s, p) = \mathcal{H}(s, p, a^*) \text{ with } \frac{\partial \mathcal{H}}{\partial a}(s, p, a^*) = 0 \qquad (7)$$

The stationarity condition $\frac{\partial \mathcal{H}}{\partial a}(s, p, a^*) = 0$ ensures that the optimal action can be implicitly represented by the pair $(s, p)$, allowing us to reformulate the optimal path solely in terms of these dual variables. This condition effectively decouples the explicit dependency on actions by encoding them through the adjoint variable $p$. In this setting, the canonical state-action pair $(s, a)$ is replaced by the extended state $(s, p)$, with the action recovered as a function $a = P(s, p)$ that solves the stationarity condition. Substituting this back yields the reduced Hamiltonian $\hbar(s, p) = p^\top f(s, P(s, p)) - r(s, P(s, p))$. Here, the influence of the reward function $r$ is now captured through the reduced Hamiltonian. We couple state and adjoint variables into the composite vector $x = (s, p)$ with dimension $d = d_\mathcal{S} + d_\mathcal{A}$ (sum of state and action space's dimensions). The **differential dual system** can now be written as:

$$\dot{x} = S\nabla\hbar(x) \qquad (8)$$

where $S$ is the canonical sympletic matrix $\begin{bmatrix} 0 & I \\ -I & 0 \end{bmatrix}$. This formulation encodes the evolution of the system through a Hamiltonian gradient flow in phase space, which serves as the foundation for our policy learning formulation. By discretizing the differential system, we arrive at the update rule:

$$x_{n+1} = x_n + \Delta_t S\nabla\hbar(x_n) := G(x_n) \qquad (9)$$

where $\Delta_t$ is the discretization time step. $G$ is the dynamics operator dictating the evolution of the policy-induced trajectory. From a learning perspective, we aim to discover an operator $G$ such that successive applications generate the optimal trajectory $x, G(x), G^{(2)}(x), \cdots$, where $G^{(k)}$ denotes the $k$-fold composition of $G$: $G^{(k)}(x) = G(G(\cdots G(x) \cdots))$.

To approximate $\hbar(x)$, we introduce a learnable *score function* $g(x) \approx \hbar(x)$, which plays the role of a surrogate reward landscape defined over the extended space $(x = (s, p))$. This reparameterization allows us to shift the learning objective toward trajectory-consistent updates. Altogether, this approximation process suggests that the original reinforcement learning problem, when viewed through the lens of continuous-time optimal control and its differential dual, can be reformulated as

an abstract policy optimization problem, denoted $\mathcal{D}$. The goal in $\mathcal{D}$ is to learn the optimal dynamics operator $G : \Omega \rightarrow \Omega$ that governs the optimal trajectory:

$$x_0 = x \sim \rho_0, \quad x_1 = G(x_0) = G(x), \tag{10}$$

$$x_2 = G(x_1) = G^{(2)}(x), \cdots, x_{H-1} = G^{(H-1)}(x) \tag{11}$$

Here $\Omega$ is a compact domain in $\mathbb{R}^d$, and $H$ is the number of steps in an episode. Moreover, $\rho_0$ is the distribution of the starting point $x_0$. In this work, we assume that $\rho_0$ is a continuous function. Performing an interaction with an environment $\mathcal{B}$ (see Definition 2.1), we learn a policy $G_\theta$ parameterized by $\theta$ that approximates $G$.

**Definition 2.1.** An environment $\mathcal{B}$ with respect to an adversarial distribution $\rho_0$ and a score function $g$ is a black-box system that allows you to check the quality of a policy $G_\theta$. More concretely, $\mathcal{B}$ inputs a policy $G_\theta$ and outputs the trajectories $(G_\theta^{(k)}(x))_{k=0}^{H-1}$, and their associated scores $(g(G_\theta^{(k)}(x)))_{k=0}^{H-1}$ for a sample $x$ from distribution $\rho_0$.

The function $g$ serves as a score surrogate that allows us to evaluate and update the current policy $G_\theta$. From the differential dual formulation (Equation (9)), we obtain a first-order relationship:

$$G = \mathrm{Id} + \Delta_t S \nabla g, \tag{12}$$

where $\mathrm{Id}$ is the identity operator, $\Delta_t$ is again the time step, and $S$ is the symplectic matrix. This formulation implicitly encodes a physics prior through the symplectic form, while still allowing flexibility for data-driven learning. As such, even though the original RL problem does not explicitly enforce physical constraints, the differential structure induces an implicit bias toward trajectory-consistent behavior, making it applicable to physics-based dynamical systems.

## 2.2 Differential policy optimization (dfPO) algorithm

Differential policy optimization (dfPO) (see Algorithm 1) is a "time-expansion (Dijkstra-like)" algorithm that iteratively uses appropriate on-policy samples for policy updates to ensure an increase in policy quality over each iteration. This algorithm has a similar idea to the trust region policy optimization (TRPO) [26]. However, because of our differential approach that focuses on pointwise estimates, dfPO becomes much simpler and easier to implement/optimize in practice, compared to TRPO and other RL counterparts.

## 2.3 Application to scientific computing

We demonstrate how Differential RL naturally applies to a broad class of scientific-computing problems by instantiating the abstract formulation $\mathcal{D}$ in three representative domains with energy-based objectives. Such objectives can either be potential-based reward structure of the form $r(s, a) = -\mathcal{F}(s)$ or an energy-regularized variant $r(s, a) = \frac{1}{2}\|a\|^2 - \mathcal{F}(s)$. Here, $\mathcal{F}(s)$ denotes a task-specific potential or cost functional.

The agent's objective is to reach low-energy states while minimizing control effort. Although the system dynamics can be simplified, the complexity of the task is fully encapsulated in $\mathcal{F}$. In many scientific settings, $\mathcal{F}$ is only accessible via simulation, lacks a closed-form expression, and cannot be queried or differentiated directly. This renders model-based RL methods, which rely on explicit reward access or gradients, inapplicable. Differential RL circumvents this limitation by relying solely on scalar evaluations along actual trajectories, similar to model-free RL. However, unlike typical model-free methods, it embeds a physics-informed inductive bias through its differential structure, making it particularly suited for scientific problems. The three representative domains include:

**Surface Modeling**: This setting involves evolving surfaces optimized for geometric or physical properties. The surface is parameterized by control points (e.g., knots in a spline), and the reward is derived from physical objectives such as smoothness, curvature, or structural stress. The state $s$ encodes the control point positions, and the action $a_k$ updates them according to $s_{k+1} = s_k + \Delta_t a_k$. The cost $\mathcal{F}(s)$ evaluates the reconstructed surface $\mathcal{S}(s)$, with rewards of the form $r(s, a) = -\mathcal{F}(s)$ or $r(s, a) = \frac{1}{2}\|a\|^2 - \mathcal{F}(s)$ to penalize excessive updates.

**Grid-based Modeling**: In many PDE-constrained problems, control is applied on a coarse spatial grid, while evaluation occurs on a refined grid. The state $s$ consists of coarse-grid values, and

---

**Algorithm 1 (Main algorithm)** dfPO for a generic environment $\mathcal{B}$

---

**Input:** a generic environment $\mathcal{B}$, the number of steps per episode $H$, time step $\Delta_t$, and the number of samples $N_k$ at stage $k$ with $k \in \overline{1, H-1}$. Here $N_k$ can be chosen based on Theorem 3.2. We also assume that the hypothesis space for the policy approximator $G_{\theta_k}$ in stage $k$ is $\mathcal{H}_k$ for $k \in \overline{1, H}$.
**Output:** a neural network approximation $G_\theta$ that approximates the optimal policy $G$.

1: Initialize an empty replay memory queue $\mathcal{M}$.
2: Initialize $k = 1$ as the current stage and a random scoring function $g_{\theta_0}$. Set the initial policy $G_{\theta_0} = \mathrm{Id} + \Delta_t S \nabla g_{\theta_0}$ via automatic differentiation.
3: **repeat**
4:     Use $N_k$ starting points $\left\{ X^i \right\}_{i=1}^{N_k}$ and previous policy $G_{\theta_{k-1}}$ to query $\mathcal{B}$ to get $N_k$ sample trajectories $\left\{ G_{\theta_{k-1}}^{(j)}(X^i) \right\}_{j=0}^{H-1}$ together with their scores $\left\{ g(G_{\theta_{k-1}}^{(j)}(X^i)) \right\}_{j=0}^{H-1}$ for $i \in \overline{1, N_k}$.
5:     Add the labeled samples of the form $(x, y) = (G_{\theta_{k-1}}^{(k-1)}(X^i), g(G_{\theta_{k-1}}^{(k-1)}))$ to $\mathcal{M}$. Also add labeled samples $(x, y) = (G_{\theta_{k-1}}^{(j)}(X^i), g_{\theta_{k-1}}(G_{\theta_{k-1}}^{(j)}(X^i)))$ for $j \in \overline{1, k-2}$ and $i \in \overline{1, N_k}$ to $\mathcal{M}$. The latter addition step is to ensure that the new policy doesn't deviate from the previous policy on samples on which the previous policy already performs well.
6:     Train the neural network $g_{\theta_k} \in \mathcal{H}_k$ at stage $k$ using labeled sample from $\mathcal{M}$ with smooth $L^1$ loss function [12].
7:     Set $G_{\theta_k} = \mathrm{Id} + \Delta_t S \nabla g_{\theta_k}$ via automatic differentiation. Update $k \to k + 1$.
8: **until** $k \geq H$
9: Output $G_{\theta_{H-1}} := \mathrm{Id} + \Delta_t S \nabla g_{\theta_{H-1}}$ via automatic differentiation.

---

actions modifying them. The reward $\mathcal{F}(s)$ is implicitly defined via a fine-grid reconstruction $s_1(s)$: $\mathcal{F}(s) = \mathcal{U}(s_1(s))$ for evaluation $\mathcal{U}$ on finer grid.

**Molecular Dynamics**: State $s_t$ encodes atomic coordinates in a fixed molecular graph, with actions as vertex displacements. The energy cost $\mathcal{F}(s)$ reflects atomic interactions over edges $E$, and the objective is to reach low-energy, physically plausible configurations via $r(s, a) = -\mathcal{F}(s)$ or variants.

To analyze the dual dynamics in Equation (7), we consider the regularized reward form $r(s, a) = \frac{1}{2}\|a\|^2 - \mathcal{F}(s)$ used across the three scientific-computing settings above. In this case, the stationarity condition $\frac{\partial \mathcal{H}}{\partial a}(s, p, a^*) = 0$ implies that $p = \frac{\partial r}{\partial a}(s, a^*) = a^*$, establishing a one-to-one correspondence between the adjoint variable and the action. Substituting back, the reduced Hamiltonian becomes $\hbar(s, p) = \frac{1}{2}\|p\|^2 - r(s, p)$, showing that the dual's central term is essentially a regularized version of the original reward. Hence the score function $g(s, p)$ can be defined as $\frac{1}{2}\|p\|^2 - r(s, p)$.

## 3 Theoretical analysis

This section shows the convergence of differential policy optimization (dfPO, Algorithm 1) based on generalization pointwise estimates. We then use this result to derive regret bounds for dfPO.

### 3.1 Pointwise convergence and sample complexity

Definition 3.1 below defines the number of training samples needed to allow derivative approximation transfer. This definition is used to derive the number of samples needed for Algorithm 1.

**Definition 3.1.** Recall that $\rho_0$ is a continuous density for the starting states. Suppose that we are given a function $g : \Omega \to \mathbb{R}$, a hypothesis space $\mathcal{H}$ consists of the function $h \in \mathcal{H}$ that approximates $g$, two positive constants $\epsilon$ and $\delta$. We define the function $N(g, \mathcal{H}, \epsilon, \delta)$ to be the number of samples needed so that if we approximate $g$ by $h \in \mathcal{H}$ via $N(g, \mathcal{H}, \epsilon, \delta)$ training samples, then with probability of at least $1 - \delta$, we have the following estimate bound on two function gradients:

$$\|\nabla g(X) - \nabla h(X)\| < \epsilon \tag{13}$$

In other words, we want $N(g, \mathcal{H}, \epsilon, \delta)$ to be large enough so that the original approximation can transfer to the derivative approximation above. If no such $N(g, \mathcal{H}, \epsilon, \delta)$ exists, let $N(g, \mathcal{H}, \epsilon, \delta) = \infty$.

**Pointwise convergence.** We now state the main theorem of pointwise convergence for dfPO below.

**Theorem 3.2.** *Suppose that we are given a threshold error $\epsilon$, a probability threshold $\delta$, and a number of steps per episode $H$. Assume that $\{N_k\}_{k=1}^{H-1}$ is the sequence of numbers of samples used at each stage in Algorithm 1 (dfPO) so that:*

$$N_1 = N(g, \mathcal{H}_1, \epsilon, \delta), \tag{14}$$

$$N_k = \max\{N(g_{\theta_{k-1}}, \mathcal{H}_k, \epsilon, \delta_{k-1}/(k-1)), N(g, \mathcal{H}_k, \epsilon, \delta_{k-1}/(k-1))\} \text{ for } k \in \overline{2, H-1} \tag{15}$$

*Here $\delta_k = \delta/3^{H-k} = 3\delta_{k-1}$. We further assume that there exists a Lipschitz constant $L > 0$ such that both the true dynamics $G$ and the policy neural network approximator $G_{\theta_k}$ at step $k$ with regularized parameters have their Lipschitz constant at most $L$ for each $k \in \overline{1, H}$. Then, for a general starting point $X$, with probability at least $1 - \delta$, the following generalization bound for the trained policy $G_{\theta_k}$ holds for all $k \in \{1, 2, \cdots, H-1\}$:*

$$\mathbb{E}_X \|G_{\theta_k}^{(j)}(X) - G^{(j)}(X)\| < \frac{jL^j\epsilon}{L-1} \text{ for all } 1 \leq j \leq k \tag{16}$$

*Note that when $N_k \to \infty$, the errors approach $0$ uniformly for all $j$ given a finite terminal time $T$.*

*Proof.* The key idea is to prove a stronger statement by induction over the stage number $k$: with probability of at least $1 - \delta_k$,

$$\mathbb{E}_X \|G_{\theta_k}^{(j)}(X) - G^{(j)}(X)\| < \epsilon_j \text{ for all } 1 \leq j \leq k \tag{17}$$

Here $\epsilon_k$ and $\alpha_k$ are sequences defined in Lemma B.1 (Appendix). The inductive step relies on bounding the error using the following decomposition with 3 components:

$$\mathbb{E}_X \|G_{\theta_{k+1}}^{(k+1)}(X) - G^{(k+1)}(X)\| \leq \mathbb{E}_X \|G_{\theta_{k+1}}(G_{\theta_{k+1}}^{(k)}(X)) - G_{\theta_{k+1}}(G_{\theta_k}^{(k)}(X))\|$$

$$+ \mathbb{E}_X \|G_{\theta_{k+1}}(G_{\theta_k}^{(k)}(X)) - G(G_{\theta_k}^{(k)}(X))\| + \mathbb{E}_X \|G(G_{\theta_k}^{(k)}(X)) - G(G^{(k)}(X))\|$$

$$\leq L\alpha_k + \epsilon + L\epsilon_k = \epsilon_{k+1} \tag{18}$$

This combines the Lipschitz continuity of $G$, the supervised approximation error, and the inductive hypothesis. A complete and formal proof is provided in the Appendix (Section B). $\square$

**Sample complexity.** The general pointwise estimates for dfPO algorithm in Theorem 3.2 allow us to explicitly state the number of training episodes required for two scenarios considered in this work: one work with general neural network approximators and the other with more restricted (weakly convex and linearly bounded) difference functions:

**Corollary 3.3.** *In Algorithm 1 (dfPO), suppose we are given fixed step size and fixed number of steps per episode $H$. Further assume that for all $k \in \overline{1, H-1}$, $\mathcal{H}_k$ is the same everywhere and is the hypothesis space $\mathcal{H}$ consisting of neural network approximators with bounded weights and biases. Then with the sequence of numbers of training episodes $N_k = \mathcal{O}(\epsilon^{-(2d+4)})$, the pointwise estimates Equation (16) hold.*

**Corollary 3.4.** *Again, in Algorithm 1 (dfPO), suppose we are given fixed step size and fixed number of steps per episode $H$. Suppose $\mathcal{H}_k$ is a **special** hypothesis subspace consisting of $h \in \mathcal{H}_k$ so that $h - g$ and $h - g_{\theta_{k-1}}$ are both p-weakly convex and linearly bounded. Then with the sequence of numbers of training episodes $N_k = \mathcal{O}(\epsilon^{-6})$, we obtain the pointwise estimates Equation (16).*

Definitions of **weakly convex** and **linearly bounded**, along with proofs of Corollary 3.3 and Corollary 3.4, are provided in the Appendix. The following corollary confirms dfPO's convergence.

**Corollary 3.5.** *Note that dynamics operator $G(x)$ has the form $x + \Delta_t F(x)$, where $\Delta_t$ is the step size, and $F$ is a bounded function. In this case, even when the step size is infinitely small, Algorithm 1 (dfPO)'s training converges with reasonable numbers of training episodes.*

*Proof.* The Lipschitz constant $L$ of $G$ in this case is bounded by $1 + C\Delta_t$ for some constant $C > 0$. By scaling, WLOG, assume that for finite-time horizon problem, the terminal time $T = 1$ so that the number of steps is $m = 1/\Delta_t$. Hence, for $N_k = \mathcal{O}(1/\Delta_t^{2p})$, $\epsilon = \mathcal{O}(\Delta_t^p)$ for $p > 2$, the error bounds in Theorem 3.2 are upper-bounded by:

$$\|G_{\theta_k}^{(k)}(X) - G^{(k)}(X)\| \leq \frac{kL^k\epsilon}{L-1} \leq \frac{1}{\Delta_t}\left(1 + \frac{C}{m}\right)^m \mathcal{O}(\Delta_t^p)\frac{1}{\Delta_t} \leq e^C \mathcal{O}(\Delta_t^{p-2}) \to 0 \tag{19}$$

for $k \leq H - 1$, as $\Delta_t \to 0$. $\square$

## 3.2 Regret bound analysis

Now we give a formal definition of the **regret** and then derive two regret bounds for dfPO algorithm (Algorithm 1). Suppose $K$ episodes are used during the training process and suppose a policy $\pi^k$ is applied at the beginning of the $k$-th episode with the starting state $s^k$ (sampled from adversary distribution $\rho_0$) for $k \in \overline{1, K}$. We focus on the total number of training episodes used and assume that the number of steps $H$ is fixed. The quantity **Regret** is then defined as the following function of the number of episodes $K$:

$$\mathbf{Regret}(K) = \sum_{k=1}^{K}(V(s^k) - V_{\pi^k}(s^k)) \tag{20}$$

We now derive an upper bound on **Regret**$(K)$ defined in Equation (20):

**Corollary 3.6.** *Suppose that number of steps per episode $H$ is fixed and relatively small. If in Algorithm 1 (dfPO), the number of training samples $N_k$ have the scale of $\mathcal{O}(\epsilon^{-\mu})$, regret bound for dfPO is upper-bounded by $\mathcal{O}(K^{(\mu-1)/\mu})$. As a result, for the general case in Corollary 3.3, we obtain a regret bound of $\mathcal{O}(K^{(2d+3)/(2d+4)})$. For the special cases with restricted hypothesis space in Corollary 3.4 the regret bound is $\mathcal{O}(K^{5/6})$.*

*Proof.* For a fixed $H$, Equation (16) gives us the estimate $\mathbb{E}_X[G_{\theta_k}^{(j)}(X) - G^{(j)}(X)] \approx \mathcal{O}(\epsilon)$ for the state-action pair $X$ at step $j$ between $(N_1 + \cdots + N_{k-1} + 1)^{th}$ and $(N_1 + \cdots + N_k)^{th}$ episodes during stage $k$. Assuming a mild Lipschitz condition on reward function, the gap between the optimal reward and the reward obtained from the learned policy at step $j$ during these episodes is also approximately $\mathcal{O}(\epsilon)$. Summing these uniform bounds over all $j$ and over all episodes gives:

$$\mathbf{Regret}(K) \leq H(N_1 + \cdots + N_{H-1})C\epsilon = CHK\epsilon \tag{21}$$

Since $N_k = \mathcal{O}(\epsilon^{-\mu})$, $K = H\mathcal{O}(\epsilon^{-\mu})$. With a fixed $H$, $\epsilon = \mathcal{O}(K^{-1/\mu})$. Hence, Equation (21) leads to $\mathbf{Regret}(K) \leq K\mathcal{O}(K^{-1/\mu}) = \mathcal{O}(K^{(\mu-1)/\mu})$ as desired.

$\square$

# 4 Experiments

## 4.1 Evaluation tasks

We evaluate Differential RL on three scientific computing tasks drawn from the domains introduced in Section 2.3. For each, we explicitly define the functional cost $\mathcal{F}(s)$ and provide the relevant mathematical details below.

**Surface Modeling.** A representative surface modeling task arises in materials engineering [7], where raw materials (e.g., metals, plastics) are deformed into target configurations. Formally, an initial shape $\Gamma_0$ are deformed into the target shape $\Gamma^*$ through a shape-dependent cost functional $\mathcal{C} : \mathcal{S}(\text{shape}) \to \mathbb{R}$. The state $s$ encodes control points on the shape's 2D boundary, which are interpolated into a smooth curve $\Gamma(s)$ using cubic splines. The functional cost is then defined as $\mathcal{F}(s) := \mathcal{C}(\Gamma(s))$, where $\mathcal{C}(\Gamma) := \dfrac{\int_{\mathbb{R}^2} |\delta 1_\Gamma|\, dx}{\sqrt{\int_{\mathbb{R}^2} 1_\Gamma\, dx}}$. Here, $\delta 1_\Gamma$ denotes the distributional derivative of the indicator function $1_\Gamma$. The action $a$ incrementally updates the boundary points, and the initial shape is sampled from $\rho_0$, a distribution over random polygons.

**Grid-based Modeling.** For this domain, control is applied on a coarse spatial grid while evaluation occurs on a finer grid. The state $s$ represents values of a function $f_{\text{coarse}}$ on the coarse grid, and the action $a$ modifies these values. A bicubic spline [4] generates a fine-scale approximation $f_{\text{finer}}$ from $s$, and the cost is defined as:

$$\mathcal{F}(s) := \mathcal{C}(f_{\text{finer}}) = \frac{\int_{\text{grid}} |\delta f_{\text{finer}}|\, dx}{\sqrt{\int_{\text{grid}} f_{\text{finer}}\, dx}} \tag{22}$$

The initial coarse-grid configuration $f_{\text{coarse}}$ is sampled from a uniform distribution.

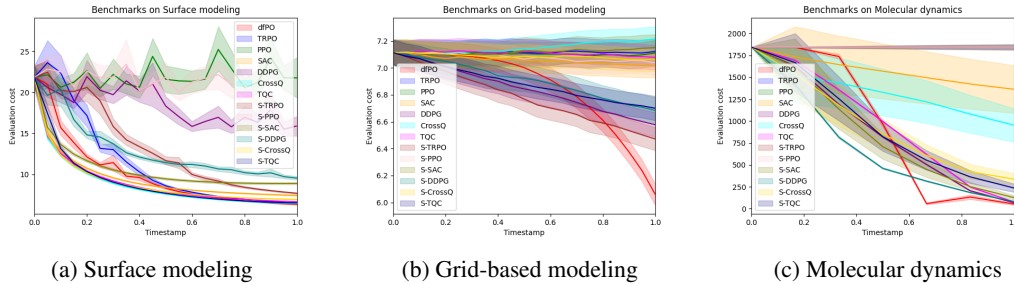

| (a) Surface modeling | (b) Grid-based modeling | (c) Molecular dynamics |

Figure 1: Evaluation costs over episodes for 13 algorithms on 3 scientific computing tasks. dfPO (red curves) consistently achieves lower costs with more optimal and physically aligned trajectories.

**Molecular Dynamics.** This task aims to guide the octa-alanine molecule to a low-energy configuration [3]. The state $s$ consists of dihedral angles $(\phi_j, \psi_j)_j$ defining the molecular conformation. Given these angles, atomic coordinates $(x_i)_{i=1}^N$ are reconstructed via a deterministic mapping $X((\phi_j, \psi_j)_j)$, based on molecular geometry. The energy functional is defined as $\mathcal{F}(s) := \mathcal{F}((\phi_j, \psi_j)_j) = U(X((\phi_j, \psi_j)_j)) = U((x_i)_{i=1}^N)$, where energy $U$ is computed via the PyRosetta package [9]. Action $a$ modifies the dihedral angle, while initial distribution $\rho_0$ is purposely chosen as a uniform distribution over small intervals to evaluate agents under limited exploration.

Beyond these examples, our framework applies to a wide range of simulation-defined objectives. Selected tasks feature sufficiently complex functionals to effectively test the proposed method. More details about our given choices of these representative tasks are given in the Appendix (Section C.6).

## 4.2 Experimental results

**Models.** We compare dfPO (Algorithm 1) against 12 baselines: 6 standard reinforcement learning (RL) algorithms and 6 energy-reshaped variants. The RL algorithms include TRPO [26] and PPO [27], two trust-region methods, with PPO widely used in LLM training due to its scalability. DDPG [19] and SAC [13] are foundational algorithms for continuous control, while Cross-Q [5] and TQC [18] offer more recent improvements in sample efficiency. For benchmarking, we denote the standard algorithms with an "S-" prefix to distinguish them from their energy-reshaped counterparts (e.g., S-PPO vs. PPO). The standard versions use the straightforward negative energy reward $r(s, a) = -\mathcal{F}(s)$, while the reshaped variants apply a time-dependent modified reward $r(s, a) = \beta^{-t}(\frac{1}{2}\|a\|^2 - \mathcal{F}(s))$. All baselines are implemented based on the Stable-Baselines3 library [24].

Table 1: Final mean evaluation costs ($\mathcal{F}(s)$ at terminal step) for all algorithms across 3 tasks. Lower values indicate better performance and correspond to higher rewards.

| | Standard Algorithms | | | | | | Reward-Shaping Variants | | | | | |
|---|---|---|---|---|---|---|---|---|---|---|---|---|
| | dfPO | S-TRPO | S-PPO | S-SAC | S-DDPG | S-CrossQ | S-TQC | TRPO | PPO | SAC | DDPG | CrossQ | TQC |
| **Surface** | 6.32 | 7.74 | 19.17 | 8.89 | 9.54 | 6.93 | 6.51 | 6.48 | 20.61 | 7.41 | 15.92 | 6.42 | 6.67 |
| **Grid** | 6.06 | 6.48 | 7.05 | 7.17 | 6.68 | 7.07 | 6.71 | 7.10 | 7.11 | 7.00 | 6.58 | 7.23 | 7.12 |
| **Mol. Dyn.** | 53.34 | 1842.30 | 1842.30 | 126.73 | 82.95 | 338.07 | 231.98 | 1842.28 | 1842.31 | 1361.31 | 68.20 | 923.90 | 76.87 |

**Train/test setup.** All models are trained under limited-sample conditions. For the first two tasks, we use 100,000 sample steps; for the third task, training is restricted to 5,000 sample steps due to the high cost of reward evaluation. Each model is evaluated over 200 test episodes with a normalized time horizon $[0, 1]$ (terminal time $T = 1$). Our model sizes are also relatively small compared to other approaches. Additional details on training samples, reward-shaping hyperparameters, and model sizes are provided in the Appendix. All experiments are conducted on an NVIDIA A100 GPU.

**Metrics.** We evaluate models based on the cost functional $\mathcal{F}$ computed over test trajectories. The objective is to achieve the lowest possible $\mathcal{F}$ values while maintaining physically plausible trajectories.

**Results.** As shown in Table 1, dfPO consistently outperforms all 12 baselines across 3 representative scientific computing tasks. No baseline (besides dfPO) dominates overall; CrossQ, TQC, DDPG,

Table 2: Hyperparameter ablations on reward-shaping algorithms.

| Dataset | dfPO | CrossQ | | | SAC | | | TQC | | |
|---|---|---|---|---|---|---|---|---|---|---|
| | orig | orig | $n_c$=10 | $n_c$=2 | orig | ent=0.05 | ent=0.2 | orig | $n_c$=10 | $n_q$=5 |
| **Surface** | **6.32** | **6.42** | 7.33 | **6.63** | 7.41 | 7.62 | 8.23 | 6.67 | 6.68 | 6.96 |
| **Grid** | **6.06** | 7.23 | 7.43 | 7.53 | **7.00** | **6.97** | 7.19 | 7.12 | 7.15 | 7.29 |
| **Mol. Dyn.** | **53.34** | 923.90 | 1247.41 | 1287.99 | 1361.31 | 1367.50 | 1386.42 | **76.87** | 98.56 | **84.36** |

| Dataset | dfPO | DDPG | | | PPO | | | TRPO | |
|---|---|---|---|---|---|---|---|---|---|
| | orig | orig | noise=OU | tau=0.01 | orig | clip=0.1 | norm=F | orig | GAE-$\lambda$=0.8 |
| **Surface** | **6.32** | 15.92 | 15.23 | 17.03 | 20.61 | 21.40 | 19.76 | **6.48** | **11.67** |
| **Grid** | **6.06** | **6.58** | 6.94 | **6.88** | 7.11 | 7.11 | 7.28 | 7.10 | 7.19 |
| **Mol. Dyn.** | **53.34** | **68.20** | 76.62 | **74.70** | 1842.31 | 1842.29 | 1842.31 | 1842.28 | 1842.33 |

and TRPO intermittently rank second or third, indicating varying strengths across domains. Notably, reward-shaped variants generally improve over their standard counterparts yet remain below dfPO. PPO underperforms across the board, likely due to its simplification of TRPO at the cost of reduced stability and weaker physics-aligned inductive bias. The evaluation-cost curves in Figure 1 show dfPO consistently exploring lower objective values with moderate variance. On the grid-based task, its advantage over baselines is clear; on surface modeling and molecular dynamics, trajectories are not always smooth but still converge to lower-energy states. dfPO's exploration pattern resembles TRPO's but attains better final values with more controlled variance. Meanwhile, SAC yields smooth curves yet fails to approach optimal values, likely due to bias from entropy regularization.

**Ablation study.** We report hyperparameter ablations for TQC—number of critics $n_c$ and quantiles $n_q$, CrossQ—number of critics $n_c$, SAC—entropy coefficient **ent**, DDPG—action noise (Ornstein–Uhlenbeck) and target-update coefficient **tau**, PPO—clip coefficient and advantage normalization, and TRPO—GAE parameter $\lambda$. dfPO uses defaults hyperparameters (learning rate 0.001, batch size 32). Reward-shaping ablation results are reported in Table 2; ablations for the standard algorithms appear in Table 5 (see Section C.3). Overall, hyperparameter variations does not substantially affect relative performance rankings, and dfPO remains robust.

**Computational complexity.** To analyze Algorithm 1, we focus on the main computational bottleneck: Step 6. In this step, the number of training updates for $g_{\theta_k}$ is proportional to the number of newly added samples to the memory buffer $M$. As shown in Corollary 3.4, this number scales as $kN_k \sim k \cdot \mathcal{O}(\epsilon^{-6})$ for each $k \in \{1, \dots, H-1\}$, where $\epsilon$ denotes the target error threshold. Therefore, the overall time complexity is $\sum_{k=1}^{H-1} k\mathcal{O}(\epsilon^{-6}) = \mathcal{O}(H^2\epsilon^{-6})$.

**Implementation link.** The complete codebase is available at https://github.com/mpnguyen2/dfPO.

# 5   Related works

**Continuous-time reinforcement learning.** While most reinforcement learning (RL) methods are formulated using Markov decision processes, control theory offers a natural continuous-time alternative [11]. Early work [31] formalized RL with a continuous-time formulation grounded in stochastic differential equations (SDEs), replacing cumulative rewards with time integrals and modeling dynamics via continuous-time Markov processes. Several subsequent works, including ours, build on this control-theoretic perspective. A related line of work proposes continuous-time policy gradient and actor-critic analogs without heavy probabilistic machinery [1, 33], but these methods also require pointwise access to rewards and their derivatives, limiting their applicability in scientific computing as discussed Section 1. Furthermore, extending SAC, TRPO, or PPO to continuous time is nontrivial: naive $Q$-function definitions collapse to the value function, eliminating action dependence and breaking advantage-based updates. Recent theory [15, 34] addresses this by redefining the $Q$-function as the limiting reward rate (expected reward per time) and linking it to the Hamiltonian (see Section 2), thereby enabling continuous-time TRPO and PPO counterparts [34].

Our work also builds on the control-theoretic formulation (simplified in Equation (4) with stochastic function $f$), but differs in two key aspects. First, we use the continuous-time formulation only as a means to derive the dual of RL: we move to continuous time mainly to construct the dual via

PMP, and then discretize the dual. Second, we define the policy over the joint space of state and adjoint variables, treating it as an operator over this extended space. This allows us to capture localized updates more naturally. We conjecture that our "$g$-function" (Section 2.1) aligns with the Hamiltonian-based $q$-function in [15], and our model corresponds to an iterative procedure refining the continuous-time advantage function within the extended state-adjoint space.

**Regret bounds.** In discrete settings, optimal $\mathcal{O}(\sqrt{K})$ regret is known (e.g., [8]), but the constants scale with the state-space size, which is intractable in continuous settings. In continuous domains, nontrivial guarantees typically require structural assumptions. Under the mild Lipschitz–MDP assumption, the minimax regret admits a lower bound $\Omega\left(K^{\frac{d+1}{d+2}}\right)$ [28], where $d$ is the joint state–action dimension. Faster rates arise with stronger smoothness: Maran et al. [21] assume $\nu$-times differentiable rewards/transitions and obtain $\mathcal{O}\left(K^{\frac{3d/2+\nu+1}{d+2(\nu+1)}}\right)$, which approaches $\mathcal{O}(\sqrt{K})$ as $\nu \to \infty$; Vakili and Olkhovskaya [30] assume kernelized rewards/transitions in an RKHS with Matérn kernel of order $m$ and show $\mathcal{O}\left(K^{\frac{d+m+1}{d+2m+1}}\right)$, again tending to $\mathcal{O}(\sqrt{K})$ as $m \to \infty$. Under comparable assumptions, our result achieves similar dimension-independent rates (see Corollary 3.6).

Our bound is significant because it is derived from *pointwise* guarantees on the per-step policy error, rather than only bounding the total cumulative regret. For a fixed horizon $H$, we show the expected policy error at each step $j$ across episode segments. Summing over steps yields the global regret (Equation (21)). These per-step guarantees are finer-grained: they show the learned policy is near-optimal at each timestep, mitigating issues like overfitting specific cumulative reward paths (e.g., reward hacking or physically inconsistent behavior). In this sense, pointwise bounds are stronger than bounding the total regret alone.

# 6   Conclusion

We propose Differential Reinforcement Learning (Differential RL), a framework that reinterprets reinforcement learning via the differential dual of continuous-time control. Unlike standard RL algorithms that rely on global value estimates, our framework offers fine-grained control updates aligned with the system's dynamics at each timestep. Differential RL also naturally introduces a Hamiltonian structure that embeds physics-informed priors, further supporting trajectory-level consistency. To implement this framework, we introduce Differential Policy Optimization (dfPO, Algorithm 1), a stage-wise algorithm that updates local movement operators along trajectories. Theoretically, we establish pointwise convergence guarantees, a property unavailable in conventional RL, and derive a regret bound of $\mathcal{O}(K^{5/6})$. Empirically, dfPO consistently outperforms standard RL baselines across three representative scientific computing domains: surface modeling, multiscale grid control, and molecular dynamics. These tasks feature complex functional objectives, physical constraints, and data scarcity, conditions under which traditional methods often struggle. Future work includes extending this framework to broader domains, investigating adaptive discretization, and further bridging the gap between optimal control theory and modern RL.

## Acknowledgments and Disclosure of Funding

This research was supported in part by a grant from the Peter O'Donnell Foundation, the Michael J. Fox Foundation, Jim Holland-Backcountry Foundation to support AI in Parkinson, and in part from a grant from the Army Research Office accomplished under Cooperative Agreement Number W911NF-19-2-0333.

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

## A  Basic algorithmic learning theory

We first introduce the results of basic learning theory on independent and identically distributed (i.i.d) samples. The proofs for all lemmas in this section can be found in [25] and can also be found in the literature on modern learning theory.

**Notations:** Throughout this section, $\mathcal{X}$ is the feature space, $\mathcal{Y}$ be the label space, $\mathcal{Z} = \mathcal{X} \times \mathcal{Y}$. Let $\mathcal{H}$ be a hypothesis space consisting of hypothesis $h : \mathcal{X} \to \mathcal{Y}$. Let $l : \mathcal{H} \times \mathcal{Z} \to \mathbb{R}_+$ be a loss function on labeled sample $z = (x, y)$ for $x \in \mathcal{X}$ and $y \in \mathcal{Y}$. Our loss function will have the following particular form: $l(h, (x, y)) = \phi(h(x), y)$ for a given associated function $\phi : \mathcal{Y} \times \mathcal{Y} \to \mathbb{R}_+$. We use capital letters to represent random variables. We also define the following sets: $\mathcal{H} \circ \{Z_1, \cdots, Z_n\} := \{(h(X_1), \cdots, h(X_n)), h \in \mathcal{H}\} \subseteq \mathbb{R}^n$, and $\mathcal{L} \circ \{Z_1, \cdots, Z_n\} := \{(l(h, Z_1), \cdots, l(h, Z_n)), h \in \mathcal{H}\} \subseteq \mathbb{R}^n$ for i.i.d samples $Z_i = (X_i, Y_i) \in \mathcal{X} \times \mathcal{Y} = \mathcal{Z}$ with label $Y_i$ with $i \in \overline{1, n}$. Also, define $e(h)$ to be the average loss over new test data $e(h) := \mathbb{E}_Z[l(h, Z)]$, and $E(h)$ the empirical loss over $n \in \mathbb{Z}_+$ i.i.d. samples $Z_1, \cdots, Z_n$: $E(h) := \frac{1}{n} \sum_{i=1}^{n} l(h, Z_i)$.

**Definition A.1.**  The Rademacher complexity of a set $\mathcal{T} \subseteq \mathbb{R}^n$ is defined as:

$$\mathbf{Rad}(\mathcal{T}) = \mathbb{E}\left[\sup_{t \in \mathcal{T}} \frac{1}{n} \sum_{i=1}^{n} B_i t_i\right] \tag{23}$$

for Bernoulli random variables $B_i \in \{-1, 1\}$.

**Lemma A.2.**  *Suppose $\phi(., y)$ is $\gamma$-Lipschitz for any $y \in \mathcal{Y}$ for some $\gamma > 0$. Then:*

$$\mathbb{E}\left[\sup_{h \in \mathcal{H}} \{e(h) - E(h)\}\right] \leq 2\,\mathbb{E}\left[\mathbf{Rad}(\mathcal{L} \circ \{Z_1, \cdots, Z_n\})\right] \tag{24}$$

$$\leq 2\gamma\,\mathbb{E}\left[\mathbf{Rad}(\mathcal{H} \circ \{Z_1, \cdots, Z_n\})\right] \tag{25}$$

**Lemma A.3.**  *For a hypothesis space $\mathcal{H}$, a loss function $l$ bounded in the interval $[0, c]$, and for $n$ i.i.d labeled samples $Z_1, \cdots, Z_n$, with probability of at least $1 - \delta$, the following bound holds:*

$$\sup_{h \in \mathcal{H}} (e(h) - E(h)) < 4\,\mathbf{Rad}(\mathcal{L} \circ \{Z_1, \cdots, Z_n\}) + c\sqrt{\frac{2\log(1/\delta)}{n}} \tag{26}$$

**Lemma A.4.**  *For hypothesis space $\mathcal{H}$ consisting of (regularized) neural network approximators with weights and biases bounded by a constant, there exists a certain constant $C_1, C_2 > 0$ so that for a set of $n$ random variables $Z_1, \cdots, Z_n$:*

$$\mathbf{Rad}(\mathcal{H} \circ \{Z_1, \cdots, Z_n\}) \leq \frac{1}{\sqrt{n}}(C_1 + C_2\sqrt{\log d}) \tag{27}$$

Throughout this paper, we assume that the optimization error can be reduced to nearly 0, so that $E(h) \approx 0$ if the hypothesis space $\mathcal{H}$ contains the function to be learned. From Lemma A.3, the average estimation error generally scales with $c\sqrt{\frac{2\log(1/\delta)}{n}}$.

## B  Proofs of theorems and corollaries in Section 3

**Proof of theorem**.  We first state the supporting Lemma B.1 and then use it to prove the main Theorem 3.2 in Section 3 regarding the pointwise estimates for dfPO algorithm.

**Lemma B.1.**  *Given $L$ and $\epsilon > 0$, define two sequences $\{\alpha_j\}_{j \geq 0}$ and $\{\epsilon_j\}_{j \geq 0}$ recursively as follows:*

$$\alpha_0 = 0 \text{ and } \alpha_j = L\alpha_{j-1} + \epsilon \tag{28}$$

$$\epsilon_1 = \epsilon \text{ and } \epsilon_{k+1} = L\alpha_k + \epsilon + L\epsilon_k \tag{29}$$

*Then for each $k$, we get:*

$$\epsilon_k \leq \frac{kL^k\epsilon}{L - 1} \tag{30}$$

*Proof.* First, $\alpha_j = (L^{j-1} + \cdots + 1)\epsilon$. Hence,

$$\frac{\epsilon_k}{L^k} \le \frac{L(L^{k-2} + \cdots + 1) + 1}{L^k}\epsilon + \frac{\epsilon_{k-1}}{L^{k-1}}$$

$$= \frac{L^k - 1}{L^k(L-1)}\epsilon + \frac{\epsilon_{k-1}}{L^{k-1}} < \frac{\epsilon}{L-1} + \frac{\epsilon_{k-1}}{L^{k-1}}$$

Hence, by simple induction, $\dfrac{\epsilon_k}{L^k} < \dfrac{(k-1)\epsilon}{L-1} + \dfrac{\epsilon_1}{L} < \dfrac{k\epsilon}{L-1}$. As a result, $\epsilon_k < \dfrac{kL^k\epsilon}{L-1}$ as desired. $\square$

Now we're ready to give a full proof for the dfPO's pointwise convergence.

**Theorem 3.2.** *Suppose that we are given a threshold error $\epsilon$, a probability threshold $\delta$, and a number of steps per episode $H$. Assume that $\{N_k\}_{k=1}^{H-1}$ is the sequence of numbers of samples used at each stage in Algorithm 1 (dfPO) so that:*

$$N_1 = N(g, \mathcal{H}_1, \epsilon, \delta), \tag{14}$$

$$N_k = \max\{N(g_{\theta_{k-1}}, \mathcal{H}_k, \epsilon, \delta_{k-1}/(k-1)), N(g, \mathcal{H}_k, \epsilon, \delta_{k-1}/(k-1))\}\, \text{for } k \in \overline{2, H-1} \tag{15}$$

*Here $\delta_k = \delta/3^{H-k} = 3\delta_{k-1}$. We further assume that there exists a Lipschitz constant $L > 0$ such that both the true dynamics $G$ and the policy neural network approximator $G_{\theta_k}$ at step $k$ with regularized parameters have their Lipschitz constant at most $L$ for each $k \in \overline{1, H}$. Then, for a general starting point $X$, with probability at least $1 - \delta$, the following generalization bound for the trained policy $G_{\theta_k}$ holds for all $k \in \{1, 2, \cdots, H-1\}$:*

$$\mathbb{E}_X \|G_{\theta_k}^{(j)}(X) - G^{(j)}(X)\| < \frac{jL^j\epsilon}{L-1} \text{ for all } 1 \le j \le k \tag{16}$$

*Note that when $N_k \to \infty$, the errors approach $0$ uniformly for all $j$ given a finite terminal time $T$.*

*Proof.* Let $\alpha_k$ and $\epsilon_k$ be two sequences associated with Lipschitz constant $L$ and threshold error $\epsilon$ as in Lemma B.1. We prove the generalization bound statement by induction on the stage number $k$ that for probability of at least $1 - \delta_k$,

$$\mathbb{E}_X \|G_{\theta_k}^{(j)}(X) - G^{(j)}(X)\| < \epsilon_j \text{ for all } 1 \le j \le k \tag{31}$$

By Lemma B.1, proving this statement also proves Theorem 3.2.

The bound for the base case $k = 1$ is due to the definition of $N_1 = N(g, \mathcal{H}_1, \epsilon, \delta)$ that allows the approximation of $g$ by $g_{\theta_1}$ transfers to (a linear transformation of) their derivatives $G$ and $G_{\theta_1}$ with error threshold $\epsilon$ and probability threshold $\delta$. Assume that the induction hypothesis is true for $k$. We prove that for a starting (random variable) point $X$, the following error holds with a probability of at least $1 - \delta_{k+1} = 1 - 3\delta_k$:

$$\mathbb{E}_X \|G_{\theta_{k+1}}^{(j)}(X) - G^{(j)}(X)\| < \epsilon_j \text{ for all } j \le k+1 \tag{32}$$

First, from induction hypothesis, with probability of at least $1 - \delta_k$:

$$\mathbb{E}_X \|G_{\theta_k}^{(j)}(X) - G^{(j)}(X)\| < \epsilon_j \text{ for all } j \le k \tag{33}$$

In stage $k+1$, all previous stages' samples up to stage $k-1$ is used for $G_{\theta_{k+1}}$. As a result, we can invoke the induction hypothesis on $k$ to yield the same error estimate for $G_{\theta_{k+1}}$ on the first $k$ sample points with probability $1 - \delta_k$:

$$\mathbb{E}_X \|G_{\theta_{k+1}}^{(j)}(X) - G^{(j)}(X)\| < \epsilon_j \text{ for all } j \le k \tag{34}$$

Recall from Algorithm 1 that $g_{\theta_{k+1}}$ is trained to approximate $g_{\theta_k}$ to ensure that the updated policy doesn't deviate too much from current policy. For $j \in \{1, \cdots, k-1\}$, $g_{\theta_{k+1}} \in \mathcal{H}_{k+1}$ approximates $g_{\theta_k}$ on $N_{k+1}$ samples of the form $G_{\theta_k}^{(j)}(X^i)$ for $i \in \{1, \cdots, N_{k+1}\}$. Since $N_{k+1} \ge N(g_{\theta_k}, \mathcal{H}_{k+1}, \epsilon, \delta_k/k)$ allows derivative approximation transfer, for probability of at least

$1 - \delta_k/k$, $\mathbb{E}_X \| G_{\theta_{k+1}}(G_{\theta_k}^{(j)}(X)) - G_{\theta_k}(G_{\theta_k}^{(j)}(X)) \| < \epsilon$. Hence, under a probability subspace $\Gamma$ with probability of at least $(1 - (k-1)\delta_k/k)$, we have:

$$\mathbb{E}_X \| G_{\theta_{k+1}}(G_{\theta_k}^{(j)}(X)) - G_{\theta_k}(G_{\theta_k}^{(j)}(X)) \| < \epsilon \tag{35}$$

for all $1 \leq j < k$

We prove by induction on $j$ that under this probability subspace $\Gamma$, we have:

$$\mathbb{E}_X \| G_{\theta_{k+1}}^{(j)}(X) - G_{\theta_k}^{(j)}(X) \| < \alpha_j \text{ for all } 1 \leq j \leq k \tag{36}$$

In fact, for the induction step, one get:

$$\begin{aligned}
\mathbb{E}_X \| G_{\theta_{k+1}}^{(j)}(X) - G_{\theta_k}^{(j)}(X) \| &\leq \mathbb{E}_X \| G_{\theta_{k+1}}(G_{\theta_{k+1}}^{(j-1)}(X)) - G_{\theta_{k+1}}(G_{\theta_k}^{(j-1)}(X)) \| \\
&\quad + \mathbb{E}_X \| G_{\theta_{k+1}}(G_{\theta_k}^{(j-1)}(X)) - G_{\theta_k}(G_{\theta_k}^{(j-1)}(X)) \| \\
&\leq L \, \mathbb{E}_X \| G_{\theta_{k+1}}^{(j-1)}(X) - G_{\theta_k}^{(j-1)}(X) \| + \epsilon \leq L\alpha_{j-1} + \epsilon = \alpha_j
\end{aligned} \tag{37}$$

Finally, we look at the approximation of $g$ by $g_{\theta_{k+1}} \in \mathcal{H}_{k+1}$ on the specific sample points $\left\{ G_{\theta_k}^{(k)}(X^i) \right\}_{i=1}^{N_{k+1}}$. Definition of $N_{k+1} \geq N(g, \mathcal{H}_{k+1}, \epsilon, \delta_k/k)$ again allow derivative approximation transfer so that with probability at least $1 - \delta_k/k$:

$$\mathbb{E}_X \| G_{\theta_{k+1}}(G_{\theta_k}^{(k)}(X)) - G(G_{\theta_k}^{(k)}(X)) \| < \epsilon \tag{38}$$

Now consider the probability subspace $\mathcal{S}$ under which 3 inequalities Equation (33), Equation (36), and Equation (38) hold. The subspace $\mathcal{S}$ has the probability measure of at least $1 - (\delta_k + (k-1)\delta_k/k + \delta_k/k) = 1 - 2\delta_k = 1 - (\delta_{k+1} - \delta_k)$, and under $\mathcal{S}$, we have:

$$\begin{aligned}
\mathbb{E}_X \| G_{\theta_{k+1}}^{(k+1)}(X) - G^{(k+1)}(X) \| &\leq \mathbb{E}_X \| G_{\theta_{k+1}}(G_{\theta_{k+1}}^{(k)}(X)) - G_{\theta_{k+1}}(G_{\theta_k}^{(k)}(X)) \| \\
&\quad + \mathbb{E}_X \| G_{\theta_{k+1}}(G_{\theta_k}^{(k)}(X)) - G(G_{\theta_k}^{(k)}(X)) \| + \mathbb{E}_X \| G(G_{\theta_k}^{(k)}(X)) - G(G^{(k)}(X)) \| \\
&\leq L \, \mathbb{E}_X \| G_{\theta_{k+1}}^{(k)}(X) - G_{\theta_k}^{(k)}(X) \| + \epsilon + L \, \mathbb{E}_X \| G_{\theta_k}^{(k)}(X) - G^{(k)}(X) \| \leq L\alpha_k + \epsilon + L\epsilon_k = \epsilon_{k+1}
\end{aligned} \tag{39}$$

Merging this inequality with probability subspace where the inequality in Equation (34) holds leads to the estimate on the final step for stage $k+1$ for the induction step. $\qquad \square$

**Proofs of corollaries**. Lemma B.3 and Lemma B.7 below estimates $N(g, \mathcal{H}, \epsilon, \delta)$ (defined in Definition 3.1) in terms of the required threshold error $\epsilon$. Such lemmas are then directly used to prove the two corollaries Corollary 3.3 and Corollary 3.4 in Section 3.

Before proving Lemma B.3 and Lemma B.7, we need the following supporting lemma.

**Lemma B.2.** *Let $\mathcal{H}$ be the hypothesis space consisting of neural network approximators with bounded weights and biases. For each $s \in \overline{1, d}$, let $\mathcal{H}_s = \{ (\nabla h)_s, h \in \mathcal{H} \} = \left\{ \frac{\partial h}{\partial x_s}, h \in \mathcal{H} \right\}$ consists of $s^{th}$ components of the gradients of elements in $\mathcal{H}$. Then the Rademacher complexity with respect to $\mathcal{H}_s$ on $n$ i.i.d random variables $Z_1, \cdots, Z_n$ scales with $\mathcal{O}(1/\sqrt{n})$:*

$$\textbf{Rad}(\mathcal{H}_s \circ \{ Z_1, \cdots, Z_n \}) = \mathcal{O}(1/\sqrt{n}) \quad \forall s \in \overline{1, d} \tag{40}$$

*Proof.* To get the Rademacher complexity bound on $\mathcal{H}_s$, we use the chain rule to express each element of $\mathcal{H}_s$ as $h_1 \cdots h_R$, where $R$ is the fixed number of layers in $\mathcal{H}$'s neural network architecture. Here each $h_i$ can be expressed as the composition of Lipschitz (activation) functions and linear functions alternatively with bounded weights and biases. Those elements $h_i$ then form another neural network hypothesis space. By invoking Lemma A.4, we obtain a bound of order $\mathcal{O}(1/\sqrt{n})$ on

individual $h_i$'s. To connect these $h_i$'s, we express the product $h_1 \cdots h_R$ as:

$$h_1 \cdots h_R = \prod_{i=1}^{R} (h_i + D - D) = \sum_{W \subseteq [R]} (-D)^{R-|W|} \prod_{i \in W} (h_i + D)$$

$$= \sum_{W \subseteq [R]} (-D)^{R-|W|} \exp \left( \sum_{i \in W} \log(h_i + D) \right)$$

Here $D$ is a constant large enough to make the $\log$ function well-defined and to make Lipschitz constant of the $\log$ function bounded above by another constant. Now we use the simple bounds on $\mathbf{Rad}(\mathcal{T} + \mathcal{T}')$ and $\mathbf{Rad}(f \circ \mathcal{T})$ for some sets $\mathcal{T}$ and $\mathcal{T}'$ and Lipschitz function $f$ to derive the Rademacher complexity bound of the same order $\mathcal{O}(1/\sqrt{n})$. $\qquad \square$

**Lemma B.3.** *Suppose that the hypothesis space $\mathcal{H}$ for approximating the target function $g : \Omega \subset \mathbb{R}^d \to \mathbb{R}^d$ consists of neural network appproximators with bounded weights and biases. In addition, assume that the function $g$ and neural network appproximators $h \in \mathcal{H}$ are continuously differentiable twice with bounded first and second derivatives by some constant $C$. One way this assumption can be satisfied is to choose activation functions that are two times continuously differentiable. Then $N(g, \mathcal{H}, \epsilon, \delta)$ is the upper bound of $\mathcal{O}(\epsilon^{-(2d+4)})$, where we only ignore the quadratic factors of $\delta$, the polynomial terms of $d$, and other logarithmic terms.*

*Proof.* Suppose we're given $\epsilon > 0$. Take $\epsilon_1 > 0$ so that $16C\epsilon_1 < \epsilon/(2d)$, and $\epsilon_2 = 0.5$. Now take $n \in \mathbb{N}$ large enough so that $C_1 \sqrt{\dfrac{\log(1/(\delta/(3d))}{n}} < \epsilon/(2d)$ for an appropriate constant $C_1$. Then take $m \in \mathbb{N}$ large enough so that $(1 - c_1 \epsilon_2 \epsilon_1^d)^m < \delta/(3n)$, where $c_1$ is an appropriate geometric constant (see the following paragraphs). Let $M = m + n$.

Train a neural network function $h \in \mathcal{H}$ to approximate the target function $g$ on $N$ samples, where $N$ is given by:

$$N = \frac{\log((6M)/\delta)}{(C\epsilon_1^2 \delta/(6M))^2} \text{ or } \sqrt{\frac{\log(1/(\delta/6M))}{N}} = C\epsilon_1^2 \frac{\delta}{6M} = N(g, \mathcal{H}, \epsilon, \delta) \qquad (41)$$

Before going to the main proof, we dissect $N$ to obtain its asymptotic rate in terms of $\epsilon$ and $d$. First of all $\epsilon_1 = \mathcal{O}(\epsilon/2d)$. Next, $n \approx \log(1/\delta)/(\epsilon/(2d))^2$, and $m \approx \log(3n/\delta)\epsilon_1^{-d}$. Hence, ignoring logarithmic terms, polynomial terms in $d$ and the quadratic factor of $\delta$, $M = m + n$ is approximately $\mathcal{O}(\epsilon^{-d})$. As a result, $N \approx \epsilon^{-(2d+4)}$.

Choose a set $S = \{X_1, \cdots, X_m, Y_1, \cdots, Y_n\}$ consisting of $M = m + n$ random samples of distribution $\rho_0$ that are independent of $g$: $m$ samples $X_1, \cdots, X_m$ and $n$ sample $Y_1, \cdots, Y_n$.

We apply Lemma A.3 to $\mathcal{H}$ with i.i.d labeled samples $Z = (X, g(X))$ and loss function $l$ with the associated $\phi(y, \hat{y}) = |y - \hat{y}|$, which is 1-Lipschitz for a fixed $\hat{y}$. In this case, from Lemma A.3, for probability of at least $1 - \delta/(6M)$, $\mathbb{E}_U[|h(U) - g(U)|] < C\epsilon_1^2 \delta/(6M)$ for the random variable $U \in S$. As a result, by Markov inequality, with probability at least $1 - \delta/(6M) - \delta/(6M) = 1 - \delta/(3M)$, $|h(U) - g(U)| < C\epsilon_1^2$ for each $U \in \{X_1, \cdots, X_m, Y_1, \cdots, Y_n\}$. Hence, there exists a probability subspace $\Gamma$ with probability at least $1 - \delta/(3M) * M = 1 - \delta/3$ so that $|h(U) - g(U)| < C\epsilon_1^2$ for all $U \in S$.

The probability that a particular sample (random variable) $X_i$ is in the hypercone with a conic angle difference of $\epsilon_2$ surrounding the direction $(\nabla h(Y) - \nabla g(Y))$ of the hyper-spherical $\epsilon_1$-circular neighborhood of $Y$ is at least $c_1 \epsilon_2 \epsilon_1^d$. Here a $\epsilon_1$-circular neighborhood here include points with radius sizes between $\epsilon_1/2$ and $\epsilon_1$. The probability that no $m$ samples is in this cone is at most $(1 - c_1 \epsilon_2 \epsilon_1^d)^m < \delta/(3n)$. As a result, on $\Gamma$ except a subspace with probability less than $\delta/(3n)$, there exists $k$ (depends on both $Y$ and $X_1, \cdots, X_m$) so that:

$$\epsilon_1/2 < \|X_k - Y\| < \epsilon_1 \qquad (42)$$
$$(\nabla h(Y) - \nabla g(Y)) \cdot (X_k - Y) > (1 - \epsilon_2)\|\nabla h(Y) - \nabla g(Y)\|\|X_k - Y\| \qquad (43)$$

Second-order Taylor expansion for $f$ and $g$ at each $Y$ yields:

$$h(X_k) = h(Y) + \nabla h(Y)(X_k - Y) + \frac{1}{2}\|X_k - Y\|^2 U_h(X_k, Y) \tag{44}$$

$$g(X_k) = g(Y) + \nabla g(Y)(X_k - Y) + \frac{1}{2}\|X_k - Y\|^2 U_g(X_k, Y) \tag{45}$$

where $U_h$ and $U_g$ are the second derivative terms of $h$ and $g$ in respectively, and are bounded by $C$. As a result:

$$
\begin{aligned}
&(1 - \epsilon_2)\|\nabla h(Y) - \nabla g(Y)\|(\epsilon_1/2) \\
&< (1 - \epsilon_2)\|\nabla h(Y) - \nabla g(Y)\|\|X_k - Y\| \\
&< (\nabla h(Y) - \nabla g(Y)) \cdot (X_k - Y) \\
&< |h(Y) - g(Y)| + |h(X_k) - g(X_k)| + 2C\|X_k - Y\|^2 \\
&< 2C\epsilon_1^2 + 2C\|X_k - Y\|^2 < 4C\epsilon_1^2
\end{aligned}
$$

Thus, $\|\nabla h(Y) - \nabla g(Y)\| < 8C\epsilon_1/(1-\epsilon_2) = 16C\epsilon_1$. Therefore, on the subspace $\Gamma_0$ with probability of at least $1 - \delta/3 - n * (\delta/(3n)) = 1 - (2\delta)/3$, $\|\nabla h(Y) - \nabla g(Y)\| < 16C\epsilon_1$ for all samples $Y \in \{Y_1, \cdots, Y_n\}$.

In order to prove that $\mathbb{E}_X\|\nabla h(X) - \nabla g(X)\| < \epsilon$ and thus finishing the proof for $N(g, \mathcal{H}, \epsilon, \delta) = \mathcal{O}(\epsilon^{-(2d+4)})$, we only need to prove bounds on individual components of $\|\nabla h(X) - \nabla g(X)\|$:

$$\mathbb{E}_X\|(\nabla h(X) - \nabla g(X))_s\| < \epsilon/d$$

where $x_s$ is the $s^{th}$ component of a vector $x \in \mathbb{R}^d$.

To this end, by Lemma B.2, we have a bound of order $\mathcal{O}(1/\sqrt{n})$ for the Rademacher complexity of the hypothesis space $\mathcal{H}_s$ for $s \in \overline{1, d}$, i.e. $\mathbf{Rad}(\mathcal{H}_s \circ \{Z_1, \cdots, Z_n\}) = \mathcal{O}(1/\sqrt{n})$ for i.i.d random variables $Z_1, \cdots, Z_n$. Hence, we can invoke Lemma A.3 on $\mathcal{H}_s$ to get:

$$
\begin{aligned}
\mathbb{E}_X\|(\nabla h(X) - \nabla g(X))_s\| &< \frac{1}{n}\sum_{i=1}^{n}\|(\nabla h(Y_i) - \nabla g(Y_i))_s\| + \epsilon/(2d) \\
&< \epsilon/(2d) + \epsilon/(2d) = \epsilon/d
\end{aligned}
$$

on $\Gamma_0$ except a set of probability of at most $\delta/(3d)$. Then we finish the proof of Lemma B.3 by summing all inequalities over $d$ components. $\qquad \square$

**Remark.** Another simpler way to achieve a similar result is to upper-bound $\|\nabla h(X) - \nabla g(X)\|$ by $\|\nabla h(Y_i) - \nabla g(Y_i)\| + 2C\|X - Y_i\|$ and use the probability subspace in which one of the $Y_i$ is close enough to $X$. However, we need a similar argument for Lemma B.7, so we moved forward with the proof approach given above.

We now state the definition of **weakly convex** and **linearly bounded** to define a more restricted hypothesis space with an improved factor in Lemma B.7.

**Definition B.4.** For $p \in \mathbb{N}$, a function $h$ is called a $p$-**weakly convex** function if for any $x \in \Omega$, there exists a sufficiently small neighborhood $U$ of $x$ so that:

$$h(y) \geq h(x) + \nabla h(x)(y - x) - C\|y - x\|^p \quad \forall y \in U \tag{46}$$

Note that any convex function is $p$-weakly convex for any $p \in \mathbb{N}$.

**Definition B.5.** A function $h$ is **linearly bounded**, if for some constant $C > 0$, and for any $x \in \Omega$, there exists a sufficiently small neighborhood $U$ of $x$ so that:

$$|h(y) - h(x)| \leq C\|\nabla h(x)\|\|y - x\| \quad \forall y \in U \tag{47}$$

Any Lipschitz function on compact domain that has non-zero derivative is linearly bounded. One such example is the function $e^{\alpha\|x\|^2}$ on domains that do not contain 0.

*Remark* B.6. Note that there are many hypothesis spaces consisting of infinitely many elements that satisfy Lemma B.7. One such hypothesis space $\mathcal{H}$ is:

$$\left\{ h(x) := \sum_{i=1}^{D} a_i e^{b_i \|x\|^2}, \quad 0 \leq a_i \leq A, \ 0 < b \leq b_i \leq B \right\} \tag{48}$$

for constant $A, B, b > 0$ and for domain $\Omega$ that doesn't contain 0. Lemma B.7 also holds for concave functions and their weak versions.

**Lemma B.7.** *Suppose that, possibly with knowledge from an outside environment or from certain policy experts, the hypothesis space $\mathcal{H}$ is reduced to a smaller family of functions of the form $h = g + h_1$, where $g$ is as in Lemma B.3, and $h_1$ is linearly bounded and p-weakly convex for some $p \geq 2d$. Then $N(g, \mathcal{H}, \epsilon, \delta)$ can be upper bound by the factor $\mathcal{O}(\epsilon^{-6})$ that is independent of the dimension $d$.*

*Proof.* Suppose we're given $\epsilon > 0$. Take $\epsilon_1 = \epsilon^{1/d} = \epsilon^\alpha$ with $\alpha = 1/d$, and set $\epsilon_2 = 1/2$. Now choose $m \in \mathbb{N}$ large enough so that $(1 - c_1 \epsilon_2 \epsilon_1^d)^m < \epsilon^2$, where $c_1$ is an appropriate geometric constant. From here, we can see that $m \approx \mathcal{O}(\epsilon_1^{-d}) = \mathcal{O}(\epsilon^{-1})$.

Choose $N \approx \mathcal{O}(\epsilon^{-6}) \in \mathbb{N}$:

$$N = \mathcal{O}\left( \frac{\log(d/\delta)}{\epsilon^6} \right) \text{ so that } \sqrt{\frac{\log(1/(\delta/d))}{N}} \approx \mathcal{O}(\epsilon^3) \tag{49}$$

Train a neural network function $h$ to approximate $g$ on $N$ samples $Y_1, \cdots, Y_N$ so that $h(Y_i) \approx g(Y_i)$ for $i \in \overline{1, N}$. We prove that this $N$ is large enough to allow derivative approximation transfer.

Choose $\beta_k = k/d$ and $\delta_k = k\delta/d$ for $k \in \overline{0, d}$. We prove by induction on $k \in \overline{0, d}$ that there exists a subspace of probability at least $1 - \delta_k$ so that $\|\nabla h(Y) - \nabla g(Y)\| < C_0 \epsilon^{\beta_k}$ for all $Y \in \{Y_1, \cdots, Y_n\}$ and for some constant $C_0$.

For $k = 0$, the bound is trivial. For the induction step from $k$ to $k + 1$, we first consider the loss function $l_k$ of the form $l_k(h, (x, y)) = l_k(h, (x, g(x))) = \phi_k(h(x), g(x))$, where $\phi_k(y, \hat{y}) = clip(|y - \hat{y}|, 0, C\epsilon^{\alpha+\beta_k})^d$. Here $clip$ denotes a clip function and, in this case, is obviously a Lipschitz function with Lipschitz constant 1. First, the loss function $l_k$ is bounded by $(C\epsilon^{\alpha+\beta_k})^d$. Now note the following simple inequality:

$$|a^d - b^d| = |a - b| \left| \sum_{k=0}^{d-1} a^k b^{d-1-k} \right| < |a - b| d(C\epsilon^{\alpha+\beta_k})^{d-1}$$

for $a, b < C\epsilon^{\alpha+\beta_k}$. The inequality shows that $\phi_k$ has the Lipschitz constant bounded by $d(C\epsilon^{\alpha+\beta_k})^{d-1}$. We are now ready to go the main step of the induction step.

Condition on $Y$, we repeat the same argument in Lemma B.3's proof to show that except for a subspace with probability less than $\epsilon^2$, there exists $j \in \overline{1, m}$ (depends on both $Y$ and $X_1, \cdots, X_m$) so that:

$$\epsilon_1/2 < \|X_j - Y\| < \epsilon_1 = \epsilon^\alpha \tag{50}$$
$$(\nabla h(Y) - \nabla g(Y)) \cdot (X_j - Y) > (1 - \epsilon_2)\|\nabla h(Y) - \nabla g(Y)\|\|X_j - Y\| \tag{51}$$

Under this subspace, because $h_1 = h - g$ is linearly bounded,

$$|h(X_j) - g(X_j)| \leq C\|\nabla h(Y) - \nabla g(Y)\|\|Y - X_j\| < C\epsilon^{\alpha+\beta_k}$$

As a result, $|h(X_j) - g(X_j)|^d = \phi_k(h(X_j), g(X_j))$. Next, since $h_1 = h - g$ is $p$-weakly convex, we have:

$$
\begin{aligned}
&(1 - \epsilon_2)\|\nabla h(Y) - \nabla g(Y)\|(\epsilon_1/2) \\
&< (1 - \epsilon_2)\|\nabla h(Y) - \nabla g(Y)\|\|X_j - Y\| \\
&\leq (\nabla h(Y) - \nabla g(Y)) \cdot (X_j - Y) \\
&\leq |h(X_j) - g(X_j)| + |h(Y) - g(Y)| + C_1\|X_j - Y\|^p \\
&< |h(Y) - g(Y)| + |h(X_j) - g(X_j)| + 2C_1(\epsilon^\alpha)^{2d} \\
&= |h(X_j) - g(X_j)| + 2C_1\epsilon^2 \\
&= \phi_k(h(X_j), g(X_j))^{1/d} + 2C_1\epsilon^2 \\
&\leq \left( \sum_{i=1}^m \phi_k(h(X_i), g(X_i)) \right)^{1/d} + 2C_1\epsilon^2
\end{aligned}
$$

By taking expectation over $X_1, \cdots, X_m$, we obtain

$$
\begin{aligned}
&(1 - \epsilon_2)\|\nabla h(Y) - \nabla g(Y)\|(\epsilon_1/2) \\
&< (1 - \epsilon^2)\left( \left( \mathbb{E}_{X_1,\cdots,X_m}\left[ \sum_{i=1}^m \phi_k(h(X_i), g(X_i)) \right] \right)^{1/d} + 2C_1\epsilon^2 \right) + C_2\epsilon^2 \\
&< \left( m\, \mathbb{E}_X[\phi_k(h(X), g(X))] \right)^{1/d} + C_3\epsilon^2 \\
&= \left( m\, \mathbb{E}_X\left[ l_k(h, (X, g(X))) \right] \right)^{1/d} + C_3\epsilon^2
\end{aligned}
$$

for appropriate constants $C_2, C_3 > 0$.

By Lemma A.2 and Lemma A.3 on the Lipschitz loss function $l_k$ bounded by $(C\epsilon^{\alpha+\beta_k})^d$ with Lipschitz constant $d(C\epsilon^{\alpha+\beta_k})^{d-1}$, with probabilty of at least $1 - \delta/d$, we can continue the sequence of upper-bounds:

$$
\begin{aligned}
&(1 - \epsilon_2)\|\nabla h(Y) - \nabla g(Y)\|(\epsilon_1/2) \\
&\leq C_4(\epsilon^{-1}(\epsilon^{\alpha+\beta_k})^{(d-1)}\epsilon^3)^{1/d} + C_3\epsilon^2 \\
&\leq C_5\epsilon^{\alpha+\beta_k+1/d} = C_5\epsilon^{\alpha+\beta_{k+1}}
\end{aligned}
$$

for appropriate constants $C_4, C_5 > 0$.

Hence, we finish the induction step because except on a subspace with probability at most $\delta_k + \delta/d = \delta_{k+1}$, the inequality for induction hypothesis at $(k + 1)$ holds. For $\beta_d = 1$, we obtain $\|\nabla h(Y) - \nabla g(Y)\| < \epsilon$ for all $Y \in \{Y_1, \cdots, Y_N\}$ with probability of at least $1 - \delta_d = 1 - \delta$. By repeating the argument in Lemma B.3's proof, we obtain the expected bound

$$
\mathbb{E}_X[\|\nabla h(Y) - \nabla g(Y)\|] < C_6\epsilon \tag{52}
$$

for an appropriate constant $C_6 > 0$. After a constant rescaling, we showed that $N(g, \mathcal{H}, \epsilon, \delta) \approx \mathcal{O}(\epsilon^{-6})$ $\qquad\square$

Together with the general pointwise estimates for dfPO algorithm in Theorem 3.2, Lemma B.3 and Lemma B.7 allow us to explicitly state the number of training episodes required for two scenarios considered in this work in Corollary 3.3 and in Corollary 3.4. Note that the proofs for these corollaries now follow trivially from Theorem 3.2, Lemma B.3, and Lemma B.7.

# C   Further experiment details

## C.1   Sample size and problem parameters

For the first two tasks, models are trained for 100,000 steps, while for the third task, training is limited to 5,000 steps due to the high computational cost of reward evaluation. For the reshaped reward $r(s,a) = \beta^{-t}(\frac{1}{2}\|a\|^2 - \mathcal{F}(s))$, we define the decay factor as $\gamma := \beta^{\Delta_t}$, where $\Delta_t$ is the step size (time step). Details on sample size (episodes and steps per episode), step size $\Delta_t$, and decay factor $\gamma$ are summarized in Table 3.

Table 3: Task-specific details.

|                    | Surface modeling | Grid-based modeling | Molecular dynamics |
|--------------------|------------------|---------------------|--------------------|
| # of episodes      | 5000             | 5000                | 800                |
| # of steps         | 20               | 20                  | 6                  |
| Step size $\Delta_t$ | 0.01           | 0.01                | 0.1                |
| Factor $\gamma$    | 0.99             | 0.81                | 0.0067             |

## C.2   Statistical analysis on benchmarking results

We perform benchmarking using 10 different random seeds, with each seed generating over 200 test episodes. In Table 4, we report the mean and variance of final functional costs across 13 algorithms. Statistical comparisons are conducted using t-tests on the seed-level means. dfPO demonstrates statistically significant improvement over all baselines in nearly all settings. The only exception is the first experiment (Surface modeling), where dfPO and CrossQ exhibit comparable performance.

Table 4: Final evaluation costs ($\mathcal{F}(s)$ at terminal step, mean $\pm$ std) from 13 different algorithms for 3 tasks from 10 different seeds.

|          | Surface modeling       | Grid-based modeling    | Molecular dynamics        |
|----------|------------------------|------------------------|---------------------------|
| dfPO     | **6.296 ± 0.048**      | **6.046 ± 0.083**      | **53.352 ± 0.055**        |
| TRPO     | 6.470 ± 0.021          | 7.160 ± 0.113          | 1842.300 ± 0.007          |
| PPO      | 20.577 ± 2.273         | 7.155 ± 0.109          | 1842.303 ± 0.007          |
| SAC      | 7.424 ± 0.045          | 7.066 ± 0.101          | 1364.747 ± 12.683         |
| DDPG     | 15.421 ± 1.471         | **6.570 ± 0.082**      | **68.203 ± 0.001**        |
| CrossQ   | **6.365 ± 0.030**      | 7.211 ± 0.122          | 951.674 ± 15.476          |
| TQC      | 6.590 ± 0.047          | 7.120 ± 0.087          | **76.874 ± 0.001**        |
| S-TRPO   | 7.772 ± 0.085          | **6.470 ± 0.098**      | 1842.287 ± 0.014          |
| S-PPO    | 16.422 ± 1.166         | 7.064 ± 0.094          | 1842.304 ± 0.009          |
| S-SAC    | 8.776 ± 0.107          | 7.209 ± 0.126          | 126.397 ± 1.315           |
| S-DDPG   | 9.503 ± 0.210          | 6.642 ± 0.124          | 82.946 ± 0.001            |
| S-CrossQ | 6.830 ± 0.076          | 7.028 ± 0.118          | 338.120 ± 8.642           |
| S-TQC    | **6.468 ± 0.026**      | 6.716 ± 0.099          | 233.944 ± 2.966           |

## C.3   Additional ablation study

In the main paper, we reported ablations for the reward-shaped variants in Table 2; here we present the corresponding results for the standard RL algorithms in Table 5 below.

## C.4   Training time and memory usage

Approximate model sizes are given in Table 6; our networks are small, so memory overhead is low and only slightly above PPO/TRPO. Approximate per-task wall-clock times are listed in Table 7 and are comparable across tasks.

Table 5: Hyperparameter ablations on standard (S-) algorithms.

| | dfPO | S-CrossQ | | | S-SAC | | | S-TQC | | |
|---|---|---|---|---|---|---|---|---|---|---|
| **Dataset** | **orig** | **orig** | **$n_c$=10** | **$n_c$=2** | **orig** | **ent=0.05** | **ent=0.2** | **orig** | **$n_c$=10** | **$n_q$=5** |
| **Surface** | 6.32 | 6.93 | 7.22 | 19.42 | 8.89 | 8.71 | 9.79 | 6.51 | 8.65 | 6.61 |
| **Grid** | 6.06 | 7.07 | 7.21 | 7.15 | 7.17 | 7.90 | 7.21 | 6.71 | 7.00 | 7.12 |
| **Mol. Dyn.** | 53.34 | 338.07 | 593.53 | 1213.82 | 126.73 | 210.94 | 523.92 | 231.98 | 270.12 | 668.10 |

| | dfPO | S-DDPG | | | S-PPO | | | S-TRPO | |
|---|---|---|---|---|---|---|---|---|---|
| **Dataset** | **orig** | **orig** | **noise=OU** | **tau=0.01** | **orig** | **clip=0.1** | **norm=F** | **orig** | **GAE-$\lambda$=0.8** |
| **Surface** | 6.32 | 9.54 | 18.63 | 11.42 | 19.17 | 19.86 | 24.97 | 7.74 | 15.41 |
| **Grid** | 6.06 | 6.68 | 6.98 | 6.95 | 7.05 | 7.14 | 7.21 | 6.48 | 6.88 |
| **Mol. Dyn.** | 53.34 | 82.95 | 90.64 | 83.74 | 1842.30 | 1842.33 | 1842.31 | 1842.30 | 1842.28 |

Table 6: Model sizes (in MB) for 13 algorithms across tasks.

| | Surface modeling | Grid-based modeling | Molecular dynamics |
|---|---|---|---|
| dfPO | 0.17 | 0.66 | 0.17 |
| TRPO | 0.06 | 0.37 | 0.06 |
| PPO | 0.08 | 0.62 | 0.08 |
| SAC | 0.25 | 2.86 | 0.25 |
| DDPG | 4.09 | 5.19 | 4.09 |
| CrossQ | 0.27 | 2.37 | 0.27 |
| TQC | 0.57 | 6.45 | 0.57 |
| S-TRPO | 0.06 | 0.37 | 0.06 |
| S-PPO | 0.08 | 0.62 | 0.08 |
| S-SAC | 0.25 | 2.86 | 0.25 |
| S-DDPG | 4.09 | 5.19 | 4.09 |
| S-CrossQ | 0.27 | 2.37 | 0.27 |
| S-TQC | 0.57 | 6.45 | 0.57 |

## C.5 Evaluation on standard RL tasks

We evaluate on continuous-state, continuous-action versions of **Pendulum**, **Mountain Car**, and **CartPole** using Gym. For **Mountain Car**, we use reward function $R = 100\,\sigma\big(20(\text{position} - 0.45)\big) - 0.1\,\text{action}[0]^2$, where $\sigma$ denotes the sigmoid. For **CartPole**, $R = \text{upright} \cdot \text{centered} \cdot \text{stable}$ with $\text{upright} = 2\sigma\big(-5\,|\theta/\theta_{\text{thresh}}|\big)$, $\text{centered} = 2\sigma\big(-2\,|x/x_{\text{thresh}}|\big)$, and $\text{stable} = 2\sigma\big(-0.5(\dot{x}^2 + \dot{\theta}^2)\big)$. Rewards lie in $[0, 1]$ and attain 1 only at $\theta = x = \dot{\theta} = \dot{x} = 0$; thus moderate reward values (e.g., $\approx 0.15$) can still indicate acceptable control within thresholds. We adopt continuous rewards to align with our continuous-time assumptions. Results in Table 8 report episode rewards.

Our method performs reasonably on these standard tasks. Additionally, in the main paper, dfPO shows strong performance in scientific computing tasks, where optimization over structured geometric spaces, coarse-to-fine grid discretizations, and molecular energy landscapes better reflect real-world modeling with complex functionals.

## C.6 Explanation on the choices of representative tasks

In this section, we justify our choice of three evaluation tasks that capture scientific-computing settings where physics and sample efficiency are essential. Our aim is to develop reinforcement learning methods for settings where data are expensive to simulate and physical consistency is critical, with a focus on scientific-computing applications. Motivated by this, we identify three representative, foundational task types:

**Surface Modeling, control over geometries.** At the level of an individual object, many scientific computing problems involve modifying the geometry of a structure to achieve desired physical properties. A standard example is the design of an airfoil (e.g., an aircraft wing), where the goal is to optimize its surface shape over time to minimize drag or maximize lift under aerodynamic flow.

Table 7: Approximate training time (in hours) for each algorithm.

| Algorithm | PPO | TRPO | SAC | DDPG | TQC | CrossQ | dfPO |
|---|---|---|---|---|---|---|---|
| **Train time (hrs)** | 0.3 | 0.6 | 1.0 | 1.2 | 2.0 | 2.0 | 1.0 |

Table 8: Episode rewards on continuous-state/action classic-control tasks.

| Task | PPO | TRPO | DDPG | SAC | TQC | CrossQ | dfPO |
|---|---|---|---|---|---|---|---|
| **Pendulum** | -0.0213 | **-0.0011** | -0.0063 | -0.0054 | -0.0047 | **-0.0045** | **-0.0042** |
| **Mountain Car** | 58.5273 | 60.1217 | 55.3489 | **60.7268** | **70.5280** | **63.2175** | 59.0146 |
| **CartPole** | 0.0903 | **0.1204** | 0.1151 | 0.1130 | 0.0527 | **0.1241** | **0.1352** |

These surfaces are often altered through a set of control points, and the reward is derived from a functional measuring aerodynamic performance. Similarly, in structural engineering, surfaces can be automatically adjusted to improve stability against external disturbances, such as seismic vibrations. Additionally, in materials processing, time-varying surface optimization is used to control mechanical or thermal properties, like stress distributions and heat dissipation, during the manufacturing of advanced materials. Our surface modeling task captures this family of problems by enabling control over geometries.

**Grid-Based Modeling, control under PDE constraints.** When moving beyond individual geometries to macro-scale physical systems, we typically encounter phenomena modeled by controlled partial differential equations (PDEs). These PDEs capture time-evolving quantities such as temperature, pressure, or concentration fields in space. For instance, the heat equation $\frac{du}{dt} = \Delta u + f$ models temperature evolution, where $u$ is the temperature field and $f$ is a control input. An important application is data center temperature control, where $f$ can represent electricity supplied to cooling elements, and the goal is to keep the temperature stable while optimizing the energy budget. Similar examples range from smart HVAC systems to industrial furnace regulation. Most, if not all, physical phenomena fall under this category and are represented by classical PDEs such as advection–diffusion equations, wave equations, reaction–diffusion systems, and elasticity equations. In practical computational settings, solving such PDEs often requires spatial discretization, typically using a grid-based approximation. Due to computational constraints, control actions are applied on a coarser grid, while the underlying physical evaluation (i.e., computing the reward) is carried out on a finer grid. Our grid-based task precisely reflects this multiscale setting: it requires learning control policies that operate on a coarse discretization but are evaluated through a fine-grid reconstruction.

**Molecular Dynamics.** At a much smaller atomic scale, such as those in molecular or biological systems, physical processes are often not well-described by a single PDE. Instead, one must work directly with the atomic structures, whose interactions are governed by complex, often nonlocal, energy-based potentials. This motivates our third category of molecular dynamics. One example is understanding how virus capsids optimally change over time under therapeutic molecular interactions. This is important for designing more effective treatments.

In summary, our three evaluation tasks correspond to core abstractions in scientific computing. As summarized in the main paper, these include:

- Optimization over geometric surfaces.
- Grid-based modeling with controlled PDEs.
- Molecular dynamics in atomistic systems.

### C.7 Scientific-computing tasks where re-planning assumptions fail

In many controlled PDEs, interesting dynamics concentrate near specific times—for example, sharp transients or blow-up behavior where $u(t) \to \infty$ as $t \to t^\star$. To resolve such phenomena, practical solvers avoid a uniform time grid and instead place time points $\{t_i\}_{i=0}^{N}$ that *cluster* near the event (often geometrically), so $t_{i+1} - t_i$ shrinks rapidly as $t_i \to t^\star$. Consider a black-box solver for such

controlled PDEs:

$$F(u_t, u, \nabla u, \nabla^2 u, f) = 0, \tag{53}$$

with a fine-grid state $x_i$ and a coarse control $f_i$. A single forward discretization step can be written as:

$$x_{i+1} = A_i\, x_i + G_i\, f_i + r_i, \tag{54}$$

where $A_i$ is the high-dimensional propagator determined by the local step size and discretization, while $G_i f_i$ injects a low-rank control effect (rank $\ll \dim x_i$), and $r_i$ is a known source/residual. Under adaptivity, the operators $(A_i)$ vary with $i$ and generally do not commute. Define the *prefix* propagators from the initial time:

$$Q_0 := I, \qquad Q_{i+1} := A_i\, Q_i. \tag{55}$$

For rollouts from $t_0$, the full-rank computations are concentrated in evaluating $Q_{i+1}x_0$, while control/source contributions are less expensive due to their low-rank structures. To keep computational cost feasible, the solver can cache low-rank surrogates of the prefixes $Q_{i+1} \approx C_{i+1}D_{i+1}$ for low-rank matrices $C_{i+1}$ and $D_{i+1}$, enabling fast computations from the initial time.

A mid-trajectory restart at an arbitrary time $t_k$ requires the full suffix operator $S_{k \to n} := A_{n-1} \cdots A_k$. Making restart practical would therefore require constructing and maintaining low-rank approximations for *every* suffix $S_{k \to n}$ across many $k$. In a black-box environment, either these suffix maps are unavailable, or storing and updating them would exceed compute and memory budgets. Thus, the re-planning assumption demands capabilities beyond a black-box solver and fails in this setting.

## D  Hamiltonian differential dual approach

### D.1  Physics intuition

Lagrangian mechanics is a reformulation of classical dynamics that expresses motion in terms of *energies* and *generalized coordinates*. Where Newtonian mechanics emphasizes forces and constraints, the Lagrangian view encodes dynamics through the *principle of stationary action*, from which the familiar conservation laws emerge. Specifically, each admissible path $s : [0, T] \to \mathbb{R}^d$ through space–time carries a scalar "action". The physical path is the one that renders this action stationary (often a minimum) under perturbations that fix the endpoints. Formally, with Lagrangian $\mathcal{L}(s, \dot{s}, t)$, the action functional $\mathcal{S}$ is defined as an indefinite integral:

$$\mathcal{S} = \int \mathcal{L}(s, \dot{s}, t)dt \tag{56}$$

and stationarity of $\mathcal{S}$ yields the Euler–Lagrange equation that governs a physical path:

$$\frac{\partial \mathcal{L}(s, \dot{s}, t)}{\partial s} = \frac{d}{dt}\frac{\partial \mathcal{L}(s, \dot{s}, t)}{\partial \dot{s}} \tag{57}$$

A canonical example is $\mathcal{L}(s, \dot{s}, t) = \frac{1}{2}m\|\dot{s}\|^2 - \mathcal{V}(s)$ (kinetic minus potential energy). Then $\partial \mathcal{L}/\partial \dot{s} = m\dot{s}$ and $\partial \mathcal{L}/\partial s = -\nabla \mathcal{V}(s)$, so that Equation (57) reduces to

$$-\nabla \mathcal{V}(s) = \frac{d}{dt}(m\dot{s}) = m\ddot{s}, \tag{58}$$

which is precisely the Newton's second law $F = m\ddot{s}$ with the force $F$ being minus gradient of the potential energy.

**Optimal control (continuous-time RL) perspective.** The Lagrangian formulation can be viewed as a special case of optimal control by identifying the control with velocity, $a \equiv \dot{s}$, and adopting the special dynamics $f(s, a) = a$. Consider the corresponding value function:

$$V(s, t) := \max_{a(\cdot)} \int_t^T \big( - \mathcal{L}(w(u), a(u), u) \big) du \quad \text{s.t.} \quad \dot{w}(u) = a(u), w(t) = s. \tag{59}$$

The Hamilton–Jacobi–Bellman (HJB) equation [11] then reads

$$\frac{\partial V(s, t)}{\partial t} + \max_a \left( \frac{\partial V(s, t)}{\partial s} f(s, a) - \mathcal{L}(s, a) \right) = 0 \tag{60}$$

Defining the Hamiltonian $\mathcal{H}(s, a, t) := \dfrac{\partial V(s,t)}{\partial s} f(s, a) - \mathcal{L}(s, a)$, we recover Equation (5) with adjoint (costate) $p = \partial V/\partial s$. Optimality requires the first-order condition $\dfrac{\partial \mathcal{H}}{\partial a} = 0$, which yields $\dfrac{\partial \mathcal{L}}{\partial \dot{s}} = \dfrac{\partial V}{\partial s}$, and substituting the maximizing control $a^*(s, t) = \dot{s}$ into Equation (60) gives $\dfrac{\partial V}{\partial t} = \mathcal{L} - \dfrac{\partial V}{\partial s} f = \mathcal{L} - \dfrac{\partial V}{\partial s} \dot{s}$.

Differentiate the identity $\dfrac{\partial \mathcal{L}}{\partial \dot{s}} = \dfrac{\partial V}{\partial s}$ along the optimal trajectory yields:

$$
\begin{aligned}
\frac{d}{dt} \frac{\partial \mathcal{L}}{\partial \dot{s}} = \frac{d}{dt} \frac{\partial V}{\partial s} &= \frac{\partial}{\partial t} \frac{\partial V}{\partial s} + \frac{\partial^2 V}{\partial s^2} \dot{s} \\
&= \frac{\partial}{\partial s} \frac{\partial V}{\partial t} + \frac{\partial^2 V}{\partial s^2} \dot{s} \\
&= \frac{\partial}{\partial s} \left( \mathcal{L} - \frac{\partial V}{\partial s} \dot{s} \right) + \frac{\partial^2 V}{\partial s^2} \dot{s} \\
&= \frac{\partial \mathcal{L}}{\partial s} - \dot{s} \frac{\partial^2 V}{\partial s^2} + \frac{\partial^2 V}{\partial s^2} \dot{s} = \frac{\partial \mathcal{L}}{\partial s}
\end{aligned}
\tag{61}
$$

which is exactly Euler-Langrange Equation (57). Thus, the stationary action is a special optimal control problem where velocity plays the role of the control, and $\mathcal{H}$ ties value gradients to momenta.

**Hamiltonian mechanics and duality.** Hamiltonian mechanics follows from the same dual construction (see Equation (9)): with controls suppressed, the Hamiltonian $\mathcal{H}$ and the adjoint $p$ encode the dynamics via symplectic flow. In our setting, Lagrangian mechanics appears as a special case, and Hamiltonian mechanics is the corresponding dual description. The differential-learning duality we use generalizes this physics correspondence and provides the bridge to continuous-time RL.

In this section, we write $a = \dot{s}$ and occasionally suppress explicit $(s, t)$ and $(s, \dot{s}, t)$ arguments in $V$ and $\mathcal{L}$ for readability. A more rigorous derivation can also be done via the calculus of variations.

### D.2 Relation with state-action value function

We revisit the temporal-difference (TD) error $r(s, a) + V(s') - V(s)$, where the next state $s'$ follows the dynamics $s' = s + \Delta_t f(s, a)$. Using a reparameterization trick, $f$ can absorb arbitrary noise $\epsilon$ as $f = f(\cdot, \epsilon)$. With a first-order expansion and a constant step size, taking $\Delta_t = 1$ to match the discrete-time TD update, we obtain:

$$
\begin{aligned}
\epsilon_{TD} = r(s, a) + V(s') - V(s) &= r(s, a) + V(s + \Delta_t f(s, a)) - V(s) \\
&\approx r(s, a) + \Delta_t f(s, a) \frac{\partial V}{\partial s}(s) = -\mathcal{H}\left( s, -\frac{\partial V}{\partial s}(s), a \right)
\end{aligned}
\tag{62}
$$

Thus, under the one-step expansion with $\Delta_t = 1$, the TD error is exactly the Hamiltonian evaluated at the value gradient. This identifies the critic's TD signal with our control-theoretic local quantities used in the dual approach. In the continuous-time limit ($\Delta_t \to 0$), this yields an instantaneous quantity that coincides with the continuous-time $q$-function of Jia and Zhou [15]. When the dynamics are unknown, such quantities can be estimated through the drift from observed transitions: $f(s, a) \approx (s' - s)/\Delta_t$.

## E    Theoretical assumptions

We briefly justify that the linearly boundedness condition in Definition B.5 is non-restrictive. The inequality can only fail at points where $\nabla h(x) = 0$ while $h$ is not locally constant. Assume the hypothesis space consists of real-analytic functions (including polynomial functions). If $h$ is real analytic and not constant, then $g(x) := \|\nabla h(x)\|^2$ is also real analytic and not identically zero. A standard fact is that the zero set of a nontrivial analytic function has Lebesgue measure zero; hence the critical set $Z := \{x \in \Omega : \nabla h(x) = 0\}$ has measure zero [22].

Fix any compact set $\mathcal{D} \subset \Omega$ and tolerance $\eta > 0$. Since $\mu(Z \cap \mathcal{D}) = 0$, choose an open neighborhood $N_\eta \supset Z$ such that $\mu(N_\eta \cap \mathcal{D}) \leq \eta$, and define $\mathcal{D}_\eta := \mathcal{D} \setminus N_\eta$. Then $\mathcal{D}_\eta$ is compact and stays a

positive distance away from $Z$, so by continuity there exist finite constants

$$m_\eta := \inf_{x \in \mathcal{D}_\eta} \|\nabla h(x)\| > 0, \qquad L := \sup_{x \in \mathcal{D}} \|\nabla h(x)\| < \infty.$$

For any $x \in \mathcal{D}_\eta$ and any $y$ in a sufficiently small neighborhood of $x$ (so that the segment $[x, y] \subset \mathcal{D}$), the mean value theorem yields:

$$|h(y) - h(x)| \leq \sup_{z \in [x,y]} \|\nabla h(z)\| \, \|y - x\| \leq L \|y - x\| \leq \frac{L}{m_\eta} \|\nabla h(x)\| \, \|y - x\|.$$

This is exactly the desired linearly boundedness inequality on $\mathcal{D}_\eta$ with constant $C_\eta := L/m_\eta$, while the excluded region has arbitrarily small measure $\mu(\mathcal{D} \setminus \mathcal{D}_\eta) \leq \eta$. Thus the assumption holds outside an arbitrarily small exceptional set (and hence with arbitrarily high probability under any distribution absolutely continuous with respect to Lebesgue measure).

For unbounded $\Omega$, the same argument applies via a countable covering by compact sets. Let $\mathcal{D}_n = \Omega \cap \overline{B(0, n)}$ and fix $\delta > 0$. Set $\eta_n = \delta/2^n$ and apply the compact-domain result on each $\mathcal{D}_n$ to obtain the "good" sets $\mathcal{G}_n \subset \mathcal{D}_n$, on which the inequality holds (with some finite constant $C_n$) and $\mu(\mathcal{D}_n \setminus \mathcal{G}_n) \leq \eta_n$. Let $\mathcal{G} := \bigcup_{n \geq 1} \mathcal{G}_n$. By a union bound,

$$\mu(\Omega \setminus \mathcal{G}) \leq \sum_{n \geq 1} \mu(\mathcal{D}_n \setminus \mathcal{G}_n) \leq \sum_{n \geq 1} \frac{\delta}{2^n} = \delta.$$

Hence, even on unbounded domains, linearly boundedness holds outside a small exceptional set.

## F  Limitation

Our theoretical results rely on a set of assumptions stated in the corresponding theorems and lemmas, including continuity of the initial state distribution and Lipschitz regularity of the dynamics operator $G$ and score function $g$. These assumptions are standard and broadly applicable in physical systems, but they exclude certain cases, such as systems with discontinuous dynamics, which are not addressed in this work.

While our Differential RL framework is designed to be broadly applicable across scientific computing domains, our experimental evaluation focuses on three representative classes: surface modeling, grid-based modeling, and molecular dynamics. These were selected to demonstrate the generality and effectiveness of our approach in settings with complex, simulation-defined objectives. Nonetheless, our experiments do not exhaust the full spectrum of possible applications. Future work will explore extensions to other domains, including those outside scientific computing, such as computer vision, and to a wider variety of functionals within each domain.

