# OpenReview forum: "A Differential and Pointwise Control Approach to Reinforcement Learning"
_NeurIPS.cc/2025/Conference — NeurIPS 2025 poster_

### Official Review · Reviewer_HKCT · 2025-06-30

**Clarity:** 3
**Significance:** 3
**Originality:** 4
**Rating:** 5
**Confidence:** 3

**Summary:**

The authors present DPO (Differential Policy Optimization), which is a continuous generalization of RL aimed at improving performance for continuous control problems where many other RL methods deliver subpar performance. DPO is build on the Pontryagin Maximum Principle and the foundations of classical Hamiltonian mechanics and enforces a symplectic structure on the evolution of the state. This makes the algorithm particularly suited for solving scientific computing problems where the system in question can be typically modeled with Hamiltonian mechanics. It also gives the algorithm point wise convergence guarantees and enables the derivation of upper bounds on the regret. The empirical performance of DPO is evaluated on a set of scientific computing problems and delivers state-of-the-art results.

**Questions:**

1.) Why is $\rho_0$ termed an adversarial distribution?

2.) Why $L_1$ loss in Step 6 of the algorithm?

3.) Symplectic for does not really encode a physical prior but rather a particular form of evolution of the system that is consistent with the classical Hamilton equations. So the derived method in the given shape only works for classical physics systems. Also for some systems, the symplectic form alone might not be enough, e.g. fluid dynamics or quantum mechanics.
Would it be possible to do something similar for fluid dynamics or quantum mechanics by incorporating other physical priors and not only the symplectic behavior?

4.) In the surface modeling task, the performance improvement of DPO seems to much less impressive compared to the other tasks. Why is that?

5.) The examples you chose have a high mathematical similarity to the proposed algorithm, especially the reward functions and the forward dynamics already look like what the algorithm does (which is fine, right tool for right problem). But what happens when you use the algorithm on a problem with less similarity? Say with more complex forward dynamics that involve a decay on state s for example or dynamics that are not Lipschitz? How much is this going to deteriorate the performance of your algorithm?

**Ethical Concerns:**

["NO or VERY MINOR ethics concerns only"]

**Final Justification:**

The authors have addressed all my remarks and were able to answer most of my question to my satisfaction. The additional experiments again emphasize the significance of the work which is why I recommend to accept it to the venue.

**Limitations:**

Yes

**Paper Formatting Concerns:**

1.) HF and hf are impractical choices for the formulation of the problem as well as for the typesetting in the manuscript. Please change that for something with higher readability, e.g. $\mathcal{H}$ and $\mathcal{h}$. Particularly eq. (8) is difficult to read.

2.) In eq (9) and others, replace $\Delta$ with $\Delta_t$ for better readability and to avoid confusion with the Laplacian operator.

3.) The subscript notation $HF_a = \dfrac{\partial HF}{\partial a}$ can be confusing to readers. Please add a footnote to explain it, e.g. in equation (7). Also creates inconsistency with notation in Ln 168.

4.) Use latex \mathcal{O} for Landau O-notation/upper bounds

5.) Ln 32: An example for a scientific problem where recomputing is not possible would be great to include in the manuscript.

6.) Ln 256: I would refrain from using the nabla $\nabla$ symbol for a distributional derivative. Use $\delta$ instead as is common in most works.

7.) Ln 259/260 and others: Use \mathrm for the typed-out subscripts "coarse", "grid" and "finer". Do this throughout the paper.

**Quality:**

2

**Strengths And Weaknesses:**

**Strengths:**

1.) Weaknesses of current state of the art are well-explained and work presented is well justified.

2.) Mathematically well founded algorithm whose derivation is sound with the goal it intends to achieve. Proofs in the main body are kept at a reasonable length and provide enough intuition to grasp the core idea.

3.) For two scenarios, the algorithm and its theoretical bounds enable the computation of the number of necessary episodes until convergence.

4.) Good experimental evaluation and documentation of procedure.

5.) Code is included and results are reproducible.

**Weaknesses:**

1.) Lack of a Related Works section.  This work could benefit from a related works section in order to showcase its advantages and compare to other algorithms and make its results stand out more, e.g. the theoretically derived regret bound.

2.) Notation is often non-standard, inconsistent and confusing. I added my proposed changes under styles.

3.) Lack of investigation for computational requirements/cost of running the algorithm compared to others. This could potentially make the algorithm stand out further. However I assume that DPO has high computational requirements than most other algorithms it compares to due to the extensive use of Automatic Differentiation.

---

> ### Author Rebuttal · Authors · 2025-07-31
>
> We would like to thank the reviewer for the recognition of our work and for the valuable and thoughtful feedback. We address the noted weaknesses and questions below:
>
> **[W1] Lack of a Related Works section**
>
> We appreciate this suggestion. In the final revision, we will include a Related Work section before the conclusion, using the additional page provided. We will highlight our contributions, particularly the theoretically derived regret bound. Our bound is notable because it arises from pointwise guarantees on per-step policy error, rather than bounding only cumulative regret.
>
> This approach offers more fine-grained guarantees: instead of bounding just a global sum, we demonstrate that the learned policy remains near-optimal at each timestep, thereby avoiding overfitting to specific cumulative reward paths that may result in inconsistent behaviors.
>
>
> **[W2] Non-standard notations**
>
> We will revise the notations and formatting based on the reviewer’s suggestions to improve clarity and consistency. All formatting issues will be addressed in the final revision.
>
> **[W3] Lack of investigation into computational requirements/cost**
>
> We apologize for not including a more comprehensive analysis of computational requirements. Section 4.2 discusses the time complexity of our method, and Appendix C.2 includes memory storage details. Below, we provide approximate training times (similar across tasks):
>
> | Algorithm     | PPO | TRPO | SAC | DDPG | TQC | Cross-Q | DPO   |
> |---------------|-----|------|-----|------|-----|----------|-------------|
> | Train time (in hours)    | 0.3 | 0.6  | 1.0   | 1.2  | 2.0    | 2.0         | 1.0            |
>
> While the use of automatic differentiation increases cost during sample collection, the overall training time is comparable to standard methods and lower than ensemble-based approaches like TQC.
>
> **[Q1]  Why is $\rho_0$ termed an adversarial distribution?**
>
> The initial state distribution can adversely impact exploration and the resulting learned policy. We use the term "adversarial" to emphasize that even under challenging initial conditions, so long as some mild regularity holds, we can still establish convergence guarantees and regret bounds, as shown in Section 3.
>
> **[Q2] Why $L_1$ loss in Step 6 of the algorithm?**
>
> Due to the complexity of the problem settings, outlier samples may occur. We use $L_1$ loss to reduce the sensitivity of the model to these outliers, improving robustness.
>
> **[Q3] Incorporate other physical priors and not only the symplectic behavior?**
>
> Yes, this is possible. The symplectic structure arises from our formulation via classical optimal control and its connection to Hamiltonian dynamics. Generalizing to other priors, such as those relevant for fluid dynamics or quantum mechanics, is a compelling direction. However, this would require alternative theoretical tools, and we consider it a very promising area for future work.
>
> **[Q4] Surface modeling task performance**
>
> Grid-based or molecular task is more complex than the surface modeling task. For instance, grid-based problems may involve multiple topological modes, making learning and decision-making more difficult. Molecular tasks involve not only atomic geometries but also their interactions, increasing complexity. In these more challenging settings, DPO has greater opportunity to demonstrate its potential.
>
> **[Q5] Handling a more complex forward dynamics that involve a decay on state s for example or dynamics that are not Lipschitz**
>
> Our theoretical results rely on regularity assumptions. When these do not hold, theoretical guarantees may break down and performance could deteriorate. However, in practice, we find that algorithmic and computational adaptations can help mitigate such issues. We are also working toward a more general framework that performs reliably under milder assumptions.
>
> We sincerely appreciate the reviewer’s thoughtful and constructive feedback. Due to the character limit, we have provided a concise response here, but we will carefully incorporate all suggestions and address the concerns raised in our final revision. Thank you again for your valuable insights and for engaging deeply with our work.

---

> > ### Comment · Reviewer_HKCT · 2025-08-02
> >
> > I thank the reviewers for carefully considering my feedback on this work. My questions and concerns were addressed to my satisfaction and I have no further questions. I maintain my score.

---

> > > ### Author Response · Authors · 2025-08-02
> > >
> > > We're glad our response addressed your questions and concerns. We sincerely appreciate your thoughtful feedback and the insightful questions you posed, which have been valuable in refining this work and guiding future directions.

---

### Official Review · Reviewer_dPSd · 2025-07-01

**Clarity:** 3
**Significance:** 3
**Originality:** 4
**Rating:** 5
**Confidence:** 4

**Summary:**

The paper develops a novel reinforcement learning algorithm: differential policy optimisation. The algorithm shifts from a “global” formulation to a local one following physical trajectories. The authors use Pontryagin’s maximum principle to update the classical RL formulation. The authors also offer a strong theoretical background regarding the algorithm’s convergence properties. Finally, several experiments are run on three different benchmarks, where DPO outperforms other DRL algorithms.

**Questions:**

1. Did you perform any sort of ablation studies on your algorithm? On the other? How sensitive are the results to such parameters? From experience, one can “make or break” the results from a DRL algorithm for a specific environment just by tweaking some hyperparameters.
2. Would it be possible for you to extend the validation section with other standard DRL environments (non-physics related)?
3. How is DPO performing on more standard physics DRL benchmarks?


Also, on a less important note:

- In the field of DRL, DPO is already widely used for Direct Preference Optimisation (Direct Preference Optimization:
Your Language Model is Secretly a Reward Model). It would be a good idea to change it.
- I would strongly advice the authors to update the timestep notation to a standard one such as $\Delta_t$ or $\Delta t$ or $\delta t$. It will also help readibility in terms of pure differential calculus ($\Delta$ being a laplacian).
- line 126 seems like a repetition
- typo for $\mathbb{R}$ line 413

**Ethical Concerns:**

["NO or VERY MINOR ethics concerns only"]

**Final Justification:**

- The authors provide a a new RL formulation, that has both good physical foundations, and is set in a continuous domain.
- The theoretical work is useful and well demonstrated.
- Results on classical benchmarks are good (as seen in the authors response).
- While tailored for the algorithm, results on the 3 benchmarks used in the main paper are enough the prove the strong usefulness of the formulation.

However, the paper lacks clarity regarding some mathematical notations and organisation but it's should be easily changed in my opinion. Similarly, the papers lacked ablation studies and other benchmarks. The authors provided the ablations and some other benchmarks. It could be extended but it is enough for now in my opinion.

Paper should be accepted in my opinion, but the authors need to:
- update notations and typos
- add ablation studies and classical benchmark
- mention other physical benchmarks

**Limitations:**

Yes.

**Paper Formatting Concerns:**

None.

**Quality:**

4

**Strengths And Weaknesses:**

*Strengths*

- The authors find and expand a new RL formulation with good physical foundations, in my opinion.
- A strong theoretical foundation is provided for pointwise convergence and sample efficiency. The proofs in the appendix are sound.
- The authors demonstrate strong performance against already well-known DRL algorithms on three different benchmarks.
- A strong learning efficiency is also demonstrated.

*Weaknesses*

- The paper only focuses on 3 (low number) benchmarks that fit their method very well in terms of pure physical formulation.
- The paper would greatly benefit from more standardized DRL benchmarks.
- The paper lacks a proper ablation study of the hyperparameters for all algorithms.
- While I know page limits make it hard to have not too dense mathematical sections, the appendix sections could benefit from more readability and being easier to follow.

---

> ### Author Rebuttal · Authors · 2025-07-31
>
> We would like to thank the reviewer for the recognition of our work and for the valuable and thoughtful feedback. We address the noted weaknesses and questions below:
>
> **[W1] Evaluation on standard RL tasks**
>
> We have additionally evaluated our algorithm on continuous-state, continuous-action versions of three standard RL tasks: Pendulum, Mountain Car, and CartPole.
>
> For Mountain Car, we modify the reward to a continuous version: $R = 100 * \sigma(20 * (position - 0.45)) - 0.1 * action[0]^2$, where $\sigma$ denotes the sigmoid function.
>
> For CartPole, we use a continuous reward: $R = \text{upright} \cdot \text{centered} \cdot \text{stable}$, with:
> * $\text{upright} = \sigma(-5 \cdot |\theta / \theta_{\text{thresh}}|)$
> * $\text{centered} = \sigma(-2 \cdot |x / x_{\text{thresh}}|)$
> * $\text{stable} = \sigma(-0.5 \cdot (\dot{x}^2 + \dot{\theta}^2))$
>
> A reward close to 1 indicates that the agent is balanced, centered, and dynamically stable.
>
> | Task         | PPO      | TRPO     | DDPG      | SAC       | TQC       | CrossQ       | DPO    |
> |--------------|----------|----------|-----------|-----------|-----------|-----------|-----------|
> | Pendulum     | -0.0213  | **-0.0011**  | -0.0063   | -0.0054   | -0.0047   | -0.0045   |  ***-0.0042***  |
> | Mountain Car | 58.5273  | 60.1217  | 55.3489   | 60.7268   | **70.5280**   | ***63.2175***   | 59.0146   |
> | CartPole     | 0.0903   | 0.1204   | 0.1151    | 0.1130    | 0.0527    | ***0.1241***     |  **0.1352**  |
>
> Our algorithm demonstrates reasonable performance on these standard tasks. More importantly, its true strength lies in scientific computing domains, where challenges such as optimizing over structured geometric spaces, handling coarse-to-fine grid discretizations, and navigating molecular energy landscapes are more representative. These tasks involve evaluating complex functionals and reflect real-world computational modeling problems.
>
> We are currently extending our method to more complex domains, such as the Black Oil Equations in reservoir engineering, a PDE-based setting similar to our grid-based task, and the molecular problem of virus capsid deformation under therapeutic interactions. These problems are more computationally intensive and domain-specific, and will be addressed in separate works. This paper focuses on the theoretical and algorithmic foundations, which provide a principled basis for these future applications.
>
> **[W2] Hyperparameter ablation study**
>
> We thank the reviewers for pointing out the lack of an ablation study. We now provide two ablation tables examining sensitivity across key hyperparameters for benchmark algorithms.
>
> In the first table, we vary parameters:
> * CrossQ / TQC: number of critics and quantiles
> * SAC: entropy coefficient
>
> | Dataset         | DPO        | CrossQ        | CrossQ         | CrossQ         | SAC         | SAC               | SAC             | TQC         | TQC             | TQC             |
> |-----------------|------------|---------------|----------------|----------------|-------------|--------------------|------------------|-------------|------------------|------------------|
> | Ab. type   | original   | original      | n_critics=10   | n_critics=2    | original    | ent_coef=0.05      | ent_coef=0.2     | original    | n_critics=10     | n_quantiles=5    |
> | Surface   | **6.32**   | 6.42          | 7.33           | 6.63           | 7.41        | 7.62               | 8.23             | 6.67        | 6.68             | 6.96             |
> | Grid      | **6.06**   | 7.23          | 7.43           | 7.53           | 7.00        | 6.97               | 7.19             | 7.12        | 7.15             | 7.29             |
> | Mol Dyn   | **53.34**  | 923.90        | 1247.41        | 1287.99        | 1361.31    | 1367.50            | 1386.42          | 76.87       | 98.56            | 84.36            |
>
> In the second table, we vary:
> * DDPG: action noise type (Ornstein-Uhlenbeck), target update coefficient
> * PPO / TRPO: clip coefficient, Generalized Advantage Estimator parameter $\lambda$, and whether to normalize the advantage
>
> | Dataset         | DPO        | DDPG        | DDPG        | DDPG         | PPO         | PPO         | PPO                      | TRPO        | TRPO               |
> |-----------------|------------|-------------|-------------|--------------|-------------|-------------|---------------------------|-------------|---------------------|
> | Ablation type   | original   | original    | noise=OU    | tau=0.01     | original    | clip=0.1    | normalize_adv=False      | original    | GAE_lambda=0.8      |
> | Surface   | **6.32**   | 15.92       | 15.23       | 17.03        | 20.61       | 21.40       | 19.76                    | 6.48        | 11.67              |
> | Grid      | **6.06**   | 6.58        | 6.94        | 6.88         | 7.11        | 7.11        | 7.28                     | 7.10        | 7.19               |
> | Mol Dyn   | **53.34**  | 68.20       | 76.62       | 74.70        | 1842.31     | 1842.29     | 1842.31                  | 1842.28     | 1842.33            |
>
> For DPO, we use standard hyperparameters including a learning rate of 1e-3 and batch size of 32, without any special tuning. The algorithm performs well under these default settings and should not require extensive hyperparameter selection.
>
> Overall, we observe that hyperparameter variation does not significantly affect relative performance rankings. Hence, we omitted these ablations from the main text due to space constraints. We report results only for the reward-reshaped version here, but the straightforward (S-) versions, omitted due to space, shows similar behavior and robustness across tasks.
>
> **[Questions]** We have addressed all questions through the responses to the weaknesses.
>
> **[Editing/formatting errors]** We will make revisions to notations and formattings based on reviewer suggestion to improve clarity.
>
> We sincerely appreciate the reviewer's thoughtful and constructive feedback. Due to the character limit, we have provided a concise response here, but we will carefully incorporate all suggestions and address the concerns raised in our final revision. Thank you again for your valuable insights and for engaging deeply with our work.

---

> > ### Comment · Reviewer_dPSd · 2025-08-01
> >
> > Thanks for answering my questions. As long as the ablations and the remarks (especially regarding the name and the timestep notation) are added in the new version of the paper, I'll increase my score.

---

> > > ### Author Response · Authors · 2025-08-01
> > >
> > > Thank you very much for your encouraging feedback. In the revised version of the paper, we will incorporate all your suggestions and address your concerns, including the ablation studies presented above, as well as updates to the name and timestep notation for improved clarity. We sincerely appreciate your engagement and support in strengthening the paper.

---

### Official Review · Reviewer_TLm5 · 2025-07-02

**Clarity:** 3
**Significance:** 2
**Originality:** 2
**Rating:** 4
**Confidence:** 3

**Summary:**

This paper introduces Differential Reinforcement Learning, a reformulation of RL through a continuous-time control perspective using a differential dual formulation. While standard RL maximizes cumulative rewards from discrete steps, Differential RL instead maximizes the time integral of rewards. To instantiate a practical algorithm within the Differential RL framework, the authors propose Differential Policy Optimization (DPO), an algorithm that optimizes a local trajectory operator by updating pointwise behavior along a trajectory. The authors theoretically analyze DPO by establishing pointwise convergence guarantees, properties not available in standard RL, and show that DPO has a regret bound of $O(K^{5/6})$, where $K$ is the number of training episodes. DPO is evaluated on a set of three tasks spanning surface modeling, grid-based modeling, and molecular dynamics.

**Questions:**

L30: I am not sure if this “re-planning assumption” claim is valid? These planners usually evaluate trajectories using a learned model, rather than assuming oracle resets for the environment. In practice, the planned trajectories may be used for a few time steps, and then the planner may be queried again to generate new trajectories at the latest time step.

L283: 5000 sample steps is very likely too small to properly evaluate the (asymptotic) performance of the standard RL baselines.

L152-161: The forward dynamics are exactly $s_{k+1}=s_{k}+\delta a_k$ for the surface modeling and grid-based modeling tasks? Is this also true for molecular dynamics? Given these simplified dynamics, it is not clear to me why the surface modeling or grid-based modeling tasks are proper evaluations.

How does the Differential RL formulation apply to partially-observable MDPs?

Table 4: Why is the standard deviation close to 0 for some methods on the molecular dynamics task?

**Ethical Concerns:**

["NO or VERY MINOR ethics concerns only"]

**Final Justification:**

Increased score -> 4 after rebuttal, since the authors address [W1] on discussion of prior work on continuous-time RL.

**Limitations:**

Yes, Appendix D.

**Quality:**

3

**Strengths And Weaknesses:**

[S1] Continuous-time formulations of reinforcement learning is an important topic, given that standard RL algorithms are broadly developed for the discrete-time setting (such as video games), but straightforwardly applied to continuous-time settings. Studying a continuous-time formulation may provide insight into new algorithm designs that improve on sample complexity requirements.

[S2] The paper is generally well written, the differential dual formulation is clearly introduced, and the theoretical analysis in Section 3 is easy to follow.

[W1] There is no section in the main paper or appendix that discusses prior work on continuous-time formulations in RL. I am not too familiar with the prior work in continuous-time RL, but from a cursory scan of the literature: [[Ainsworth et al., 2020]](https://arxiv.org/abs/2012.06684), [[Wang et al., 2020]](https://jmlr.org/papers/v21/19-144.html), [[Yildiz et al., 2021]](https://arxiv.org/abs/2102.04764), [[Jia and Zhou, 2022]](https://arxiv.org/abs/2207.00713), [[Treven et al., 2023]](https://arxiv.org/abs/2310.19848), [[Zhao et al., 2023]](https://arxiv.org/abs/2305.18901), [[Wallace and Si., 2024]](https://www.jmlr.org/papers/v25/24-0017.html) …
The authors should conduct a thorough literature review and contextualize their differential dual formulation to existing frameworks for continuous-time RL.

[W2] DPO relies on pointwise (local) updates, and two of the evaluation tasks (surface modeling, grid-based modeling) are designed to directly update states based on actions: $s_{k+1} = s_{k} + \delta a_k$. It is difficult to evaluate potential practical applications of DPO given these simplified dynamics. The authors may consider validating DPO on more standard control benchmarks, such as double pendulum.

[W3] There is no discussion on computational cost or hyperparameter sensitivity / tuning for the proposed DPO algorithm.

- - -

I am willing to increase my score if [W1] is properly addressed.

---

> ### Author Rebuttal · Authors · 2025-07-31
>
> We would like to thank the reviewer for your interest in our work and for the valuable and thoughtful feedback. We address the noted weaknesses and questions below:
>
> **[W1] Continuous-time RL related works**
>
> We appreciate this excellent suggestion. In the final revision, we will include a dedicated Related Work section (on the additional page) to better contextualize our contributions. In particular, we will highlight how our approach compares to prior formulations in continuous-time RL, including:
>
> > **Continuous-time RL.** Most RL methods are formulated using discrete-time Markov decision processes. However, control theory [Fleming and Soner, 2006] has long offered a natural framework for modeling RL in continuous time. One of the first works to formalize this connection is [Wang et al., 2020], which introduced a continuous-time RL formulation grounded in stochastic differential equations (SDEs), replacing cumulative rewards with time integrals and modeling dynamics via continuous-time Markov processes. Several subsequent works, including ours, build on this control-theoretic perspective.
>
> > A line of work [Ainsworth et al., 2020; Yildiz et al., 2021], adopts control-theoretic formulations and proposes continuous-time analogs of policy gradient and actor-critic methods without introducing extensive probabilistic tools. Nonetheless, these approaches require direct access to the reward function and its derivatives pointwise, suffering from the same limitations in scientific computing domains as model-based RL approaches discussed in Section 1.
>
> > To extend robust methods like SAC, TRPO, or PPO to continuous time, the Q-function must be redefined so concepts like the advantage function remain meaningful. However, naive extensions of the Q-function collapse to the value function, eliminating action dependence and breaking key components like advantage-based updates. A recent novel line of theoretical work [Jia and Zhou, 2022; Zhao et al., 2023] addresses this by redefining the Q-function as a limiting ratio of expected reward to time, connecting it to the Hamiltonian (see Section 2). [Zhao et al. 2023] further reconstruct key components of TRPO and PPO, offering continuous-time counterparts of these algorithms.
>
> > Our work also builds on the control-theoretic formulation (simplified in Eq. 4 with stochastic $f$), but differs in two key aspects. First, we use the continuous-time formulation only as a means to derive the dual of RL: we move to continuous time mainly to construct the dual via PMP, and then discretize the dual. Second, we define the policy over the joint space of state and adjoint variables, treating it as an operator over this extended space. This allows us to capture localized updates more naturally. We conjecture that our "$g$-function" (Section 2.1) aligns with the Hamiltonian-based Q-function in [Jia and Zhou, 2022], and our model corresponds to an iterative procedure refining the continuous-time advantage function within the extended state-adjoint space.
>
> We agree this context was important. To keep the paper accessible, we initially only focused on general RL literature, but we will incorporate relevant continuous-time RL works in the final revision.
>
> **[W2] Evaluation on standard RL tasks**
>
> We additionally evaluate DPO on continuous-state, continuous-action versions of Pendulum, Mountain Car, and CartPole.
>
> For Mountain Car, we use a continuous reward: $R = 100 * \sigma(20 * (position - 0.45)) - 0.1 * action[0]^2$, where $\sigma$ denotes the sigmoid function.
>
> For CartPole, we use the continuous reward $R = \text{upright} \cdot \text{centered} \cdot \text{stable}$, with:
> * $\text{upright} = \sigma(-5 \cdot |\theta / \theta_{\text{thresh}}|)$
> * $\text{centered} = \sigma(-2 \cdot |x / x_{\text{thresh}}|)$
> * $\text{stable} = \sigma(-0.5 \cdot (\dot{x}^2 + \dot{\theta}^2))$
>
> Rewards close to 1 indicate the agent is balanced, centered, and stable.
>
> We report the final mean reward below (best in bold, second-best in italic):
>
> | Task         | PPO      | TRPO     | DDPG      | SAC       | TQC       | CrossQ       | DPO    |
> |--------------|----------|----------|-----------|-----------|-----------|-----------|-----------|
> | Pendulum     | -0.0213  | **-0.0011**  | -0.0063   | -0.0054   | -0.0047   | -0.0045   |  ***-0.0042***  |
> | Mountain Car | 58.5273  | 60.1217  | 55.3489   | 60.7268   | **70.5280**   | ***63.2175***   | 59.0146   |
> | CartPole     | 0.0903   | 0.1204   | 0.1151    | 0.1130    | 0.0527    | ***0.1241***     |  **0.1352**  |
>
> While DPO performs reasonably on standard tasks, its main strengths lie in scientific domains, where policies must optimize over structured spaces, handle coarse-to-fine grid, and navigate complex energy landscapes.
>
> We are extending DPO to more complex applications, such as Black Oil Equations in reservoir engineering, which is similar to our grid-based task, and molecular systems like virus capsid deformation. These settings are more computationally demanding and will be explored in future work. This paper focuses on the theoretical and algorithmic foundation to support such directions.
>
> **[W3-part 1] Hyperparameter ablation study**
>
> We now provide ablation study varying key hyperparameters:
>
> * CrossQ / TQC: number of critics and quantiles
> * SAC: entropy coefficient
> * DDPG: noise type (Ornstein-Uhlenbeck), target update
> * PPO / TRPO: clip coefficient, GAE $\lambda$, advantage normalization
>
> | Dataset         | DPO        | CrossQ        | CrossQ         | CrossQ         | SAC         | SAC               | SAC             | TQC         | TQC             | TQC             |
> |-----------------|------------|---------------|----------------|----------------|-------------|--------------------|------------------|-------------|------------------|------------------|
> | Ab. type   | original   | original      | n_critics=10   | n_critics=2    | original    | ent_coef=0.05      | ent_coef=0.2     | original    | n_critics=10     | n_quantiles=5    |
> | Surface   | **6.32**   | 6.42          | 7.33           | 6.63           | 7.41        | 7.62               | 8.23             | 6.67        | 6.68             | 6.96             |
> | Grid      | **6.06**   | 7.23          | 7.43           | 7.53           | 7.00        | 6.97               | 7.19             | 7.12        | 7.15             | 7.29             |
> | Mol Dyn   | **53.34**  | 923.90        | 1247.41        | 1287.99        | 1361.31    | 1367.50            | 1386.42          | 76.87       | 98.56            | 84.36            |
>
> Hyperparameter variations have limited impact on relative rankings. Due to character limit, we include only one representative table from the reshaped-reward setting. Other ablation tables (including those for S-variants and remaining configurations) show similar trends and are consistent with the findings presented here.
>
> DPO uses default hyperparameters (learning rate 1e-3, batch size 32) and achieves good performance without tuning.
>
> **[W3-part 2] Computational/memory cost**
>
> Section 4.2 analyzes DPO's complexity; memory details appear in Appendix C.2. Below are approximate training times (similar across tasks):
>
> | Algorithm     | PPO | TRPO | SAC | DDPG | TQC | Cross-Q | DPO    |
> |---------------|-----|------|-----|------|-----|----------|-------------|
> | Train time (in hours)    | 0.3 | 0.6  | 1.0   | 1.2  | 2.0    | 2.0         | 1.0            |
>
> **[Q1] Re-planning assumption**
>
> We assume that planners may use learned models instead of a real oracle for re-planning, but this still relies on accurately modeling rewards from arbitrary states. Achieving this requires a well-updated policy, which in turn depends on effective re-planning, creating a circular dependency.
>
> **[Q2] 5000 sample steps too small for evaluation**
>
> We agree this number may appear low compared to typical RL benchmarks. However, our molecular task is representative of real-world biophysical simulations, where reward evaluations can take minutes per step. Running 50,000 steps would require weeks of compute per trial, making research iterations infeasible. We therefore use 5,000 steps to stress-test sample efficiency under realistic constraints. For all other tasks, we use much larger budgets (e.g., 100,000 steps), ensuring that our overall evaluation remains well-balanced.
>
> **[Q3] The forward dynamics for the surface modeling and grid-based modeling tasks**
>
> The underlying surface dynamics are inherently complex and nonlinear. For RL modeling, we define control at selected locations (e.g., surface control points), which is a common approach in scientific computing. In surface modeling, for example, one may also choose to modify the magnitude of a surface normal vector, leading to different dynamics and reward characteristics. Choosing one over the other reflects a modeling decision rather than a reduction in task complexity. Our chosen task still preserves the essential challenges of optimization over structured spatial domains.
>
> **[Q4] How does the Differential RL formulation apply to partially-observable MDPs?**
>
> DPO can be extended to POMDPs using belief states. The dual formulation remains valid, but requires probabilistic estimation of latent dynamics, and this is an exciting direction for our future work.
>
> **[Q5] Table 4: Why is the standard deviation close to 0 for some methods on the molecular dynamics task?**
>
> The initial state distribution for this task is concentrated in a narrow region, and some methods converge quickly with minimal exploration. This can result in low variance across runs.
>
> We sincerely appreciate the reviewer’s thoughtful and constructive feedback. Due to the character limit, we have provided a concise response here, but we will carefully incorporate all suggestions and address the concerns raised in our final revision. Thank you again for your valuable insights and for engaging deeply with our work.

---

> > ### Comment · Reviewer_TLm5 · 2025-08-05
> >
> > > [W1]
> >
> > I have increased my score (-> 4), assuming the comparison to prior work in continuous-time RL is added to the final version.
> >
> > > [W2]
> >
> > While I appreciate these additions for more standard RL tasks (Pendulum, Mountain Car, Cartpole), these three tasks are very simple to solve. Additionally, with these custom (smooth) reward definitions, it is also unclear how well methods are performing.
> > - Does DPO fail to solve these tasks with the original reward formulations? For instance, IsaacGym Cartpole uses: $ R=1  - \theta^2 - 0.01 |\dot{x}\_{cart}| - 0.005 |\dot{x}\_{pole}| $, whereas Gym Cartpole uses an alive reward of $R=1$.
> > - For Cartpole, if a reward of 1 indicates that the task is solved, why are the final rewards after training between 0.1 - 0.15?
> >
> > > [Q1]
> >
> > My point on the ''re-planning assumption'' is whether this is actually a limitation of model-based RL (as is written in L24-L30) which learns dynamics (and reward) models, or rather just planning / shooting algorithms (that use oracle environment reset to given states).

---

> > > ### Author Response · Authors · 2025-08-06
> > >
> > > We would like to sincerely thank the reviewer for recognizing our work, for increasing the score, and for the depth of engagement throughout the review process. Your thoughtful feedback has been very helpful in improving our paper.
> > >
> > >
> > > We will include the expanded Related Work section in the final revision, as discussed in our rebuttal, to compare our method to prior frameworks in continuous-time RL.
> > >
> > > **[W2] Evaluation on standard RL tasks**
> > >
> > > We hope to clarify our choice of using continuous reward functions. While our algorithm achieved a similar and reasonable performance under the original Mountain Car reward, we opted for a continuous version to align with the theoretical work of the paper, and we applied the same to Cartpole. Our regret bounds are derived under a Lipschitz continuity assumption on the reward function. This assumption is acknowledged in the Limitations section and is consistent with common practice in the regret literature, where even stronger smoothness conditions, such as $C^k$-differentiability for $k \geq 1$, are often assumed (e.g., [Vakili and Olkhovskaya 2024; Maran et al. 2024]).
> > >
> > > Additionally, recent continuous-time RL formulations (e.g., [Jia and Zhou 2022]) define Q-functions as limiting ratios of expected reward to time, which may be ill-defined under non-smooth rewards. While we adopted continuous versions to maintain consistency with these works, the resulting rewards still remain the soft approximations of the original ones.
> > >
> > > **Clarifying Cartpole's reward**
> > >
> > > First, we apologize for a typo error in the original rebuttal, as we missed a scaling factor of 2 in the reward terms. The corrected components are:
> > > * $\text{upright} = 2\sigma(-5 \cdot |\theta / \theta_{\text{thresh}}|)$
> > > * $\text{centered} = 2\sigma(-2 \cdot |x / x_{\text{thresh}}|)$
> > > * $\text{stable} = 2\sigma(-0.5 \cdot (\dot{x}^2 + \dot{\theta}^2))$
> > >
> > > and the reward $R = \text{upright} \cdot \text{centered} \cdot \text{stable}$.
> > >
> > > Each term is upper bounded by 1 and you're correct that the reward is bounded by 1. However, the reward reaches 1 only when the pole is perfectly upright and centered with zero velocity, i.e., $\theta = x = \dot{\theta} = \dot{x} = 0$. So while the theoretical maximum reward is 1, the agent may receive a value such as 0.15 even when all variables are well within the safe threshold region (e.g., $\theta / \theta_{\text{thresh}} \in [-0.5, 0.5]$). Hence, a reward of $0.15 \approx 2\sigma(-5 * 0.5)$ is still an indicative of task done rather than failure. We originally used the Gym environment, but IsaacGym's variant that you suggested can be more suitable.
> > >
> > > We agree that these standard RL tasks are relatively simple, especially compared to our scientific computing settings.
> > >
> > > Our focus is on scientific computing domains, and in line with your suggestion, we plan to include not only standard RL tasks but also additional experiments on more challenging benchmarks in the appendix of the final revision.
> > >
> > > **[Q1] Re-planning assumption**
> > >
> > > Thank you for rephrasing this point. In lines 24–30, our intention was to highlight the limitation of (just) planning/shooting algorithms, which require resetting to arbitrary states.
> > >
> > > We fully agree that using a learned model as a synthetic oracle is a promising approach, particularly in domains like scientific computing where real samples are often limited. At the same time, modeling challenges can arise due to the complex structure of environments such as PDE-driven simulations. In a separate, more computationally intensive project involving the Black Oil Equation, we observed that limited real data made it difficult to train both dynamic and reward model with sufficient accuracy. We believe that integrating synthetic oracles with our framework could help mitigate these challenges, and we will actively explore this direction in future work.
> > >
> > > Thank you again for your thoughtful and constructive feedback. We are very grateful for your input, which helped clarify and strengthen the work. We will carefully incorporate your suggestions in the final revision.
> > >
> > > **Reference**
> > >
> > > [Vakili and Olkhovskaya 2024] Vakili, S., Olkhovskaya, J.: Kernelized reinforcement learning with order optimal regret bounds (2024)
> > >
> > > [Maran et al. 2024] Maran, D., Metelli, A.M., Papini, M., Restell, M.: No-regret reinforcement learning in smooth MDPs (2024)

---

### Official Review · Reviewer_X9Be · 2025-07-03

**Clarity:** 2
**Significance:** 3
**Originality:** 2
**Rating:** 4
**Confidence:** 2

**Summary:**

This paper proposes Differential Reinforcement Learning (Differential RL), a new framework for continuous state-action RL that emphasizes the _dynamics of the optimal trajectory_ rather than the cumulative reward. The authors introduce an algorithm called Differential Policy Optimization (DPO), which optimizes the policy in a pointwise, stage-wise manner along the trajectory. This approach induces a Hamiltonian structure that can incorporate physics priors, ensuring that learned trajectories remain physically consistent without adding explicit constraints.

**Questions:**

For the current set of evaluations, more reasonings on the selection of the specific tasks should be provided. Also more concrete examples should be given for all 3 evaluation tasks, and reasonings behind the selections of the hyperparameters.

To strengthen the paper, the authors should consider evaluating DPO on more standard RL environments or a wider variety of tasks.

**Ethical Concerns:**

["NO or VERY MINOR ethics concerns only"]

**Final Justification:**

I decide to keep my scores.

**Limitations:**

The notion of “differential RL” and the dual formulation is rooted in control theory, and readers not familiar with Hamiltonian systems or Pontryagin’s principle might struggle without additional intuitive explanation.

**Quality:**

2

**Strengths And Weaknesses:**

This paper demonstrated a noval framework that could be potentially very interesting. The authors proved pointwise convergence of DPO, meaning the policy at each stage of the trajectory converges to the optimal action for that state. By enforcing trajectory-wide optimality and embedding physics knowledge, I believe that this approach is interesting. However, it is pending testify on how the algorithms are going to perform with real-world/simulation tasks, and in the experiements section it is unclear on how many cases are tested.
All experimental domains are physics-based tasks with specific Lagrangian energy reward structures. While appropriate for showing off DPO’s strengths, these tasks are niche and structured. It remains unclear how DPO would perform on more conventional RL benchmarks.
While the $O(K^{5/6})$ regret bound is comparable to some prior continuous-RL results, it is still a fairly slow rate (sublinear but not as favorable as $O(\sqrt{K})$) and only holds under the aforementioned special conditions.

---

> ### Author Rebuttal · Authors · 2025-07-31
>
> We would like to thank the reviewer for the recognition of our work and for the valuable and thoughtful feedback. We address the noted weaknesses and questions below:
>
> **[W1] Performance on standard RL tasks and reasoning behind the selection of 3 evaluation tasks**
>
> We have additionally evaluated our algorithm on continuous-state, continuous-action versions of three standard RL tasks: Pendulum, Mountain Car, and CartPole.
>
> For Mountain Car, we modify the reward to a continuous version: $R = 100 * \sigma(20 * (position - 0.45)) - 0.1 * action[0]^2$, where $\sigma$ denotes the sigmoid function.
>
> For CartPole, we use a continuous reward: $R = \text{upright} \cdot \text{centered} \cdot \text{stable}$, with:
> * $\text{upright} = \sigma(-5 \cdot |\theta / \theta_{\text{thresh}}|)$
> * $\text{centered} = \sigma(-2 \cdot |x / x_{\text{thresh}}|)$
> * $\text{stable} = \sigma(-0.5 \cdot (\dot{x}^2 + \dot{\theta}^2))$
>
> A reward close to 1 indicates that the agent is balanced, centered, and dynamically stable.
>
> We report the final mean reward below (best in bold, second-best in italic):
>
> | Task         | PPO      | TRPO     | DDPG      | SAC       | TQC       | CrossQ       | DPO    |
> |--------------|----------|----------|-----------|-----------|-----------|-----------|-----------|
> | Pendulum     | -0.0213  | **-0.0011**  | -0.0063   | -0.0054   | -0.0047   | -0.0045   |  ***-0.0042***  |
> | Mountain Car | 58.5273  | 60.1217  | 55.3489   | 60.7268   | **70.5280**   | ***63.2175***   | 59.0146   |
> | CartPole     | 0.0903   | 0.1204   | 0.1151    | 0.1130    | 0.0527    | ***0.1241***     |  **0.1352**  |
>
> Our algorithm demonstrates reasonable performance on these standard tasks. More importantly, its true strength lies in scientific computing domains, where challenges such as optimizing over structured geometric spaces, handling coarse-to-fine grid discretizations, and navigating molecular energy landscapes are more representative. These tasks involve evaluating complex functionals and reflect real-world computational modeling problems.
>
> We are currently extending our method to more complex domains, such as the Black Oil Equations in reservoir engineering, a PDE-based setting similar to our grid-based task, and the molecular problem of virus capsid deformation under therapeutic interactions. These problems are more computationally intensive and domain-specific, and will be addressed in separate works. This paper focuses on the theoretical and algorithmic foundations, which provide a principled basis for these future applications.
>
> **[W2] Regret bound compared to previous literature**
>
> To evaluate sample efficiency, **Regret**, the gap between the cumulative reward of the learned policy and the optimal policy over $K$ episodes, is a standard metric. In discrete settings, several works (e.g., [Cai et al., 2020]) have established optimal regret bounds of $O(\sqrt{K})$. However, these bounds depend on the cardinality of the state space, which becomes intractable in continuous state-action settings.
>
> In continuous domains, regret bounds are hard to obtain without strong assumptions. A mild assumption is that of a Lipschitz MDP, under which the minimax regret has a known lower bound of $\Omega(K^{\frac{d+1}{d+2}})$ [Slivkins, 2022], where $d$ is the dimension of the joint state-action space. Within our policy optimization (DPO) framework, we prove a regret bound of similar order: $O(K^{\frac{2d+3}{2d+4}})$ (see Corollary 3.6).
>
> To improve rates, several works impose stronger smoothness assumptions. [Maran et al. 2024] assume reward and transition functions are $\nu$-times differentiable, achieving regret $O(K^{\frac{3d/2+\nu+1}{d+2(\nu+1)}})$, which recovers the optimal $O(\sqrt{K})$ as $\nu \to \infty$. [Vakili and Olkhovskaya 2024] assume kernelized transition and reward functions in a reproducing kernel Hilbert space (RKHS) with Matern kernel of order $m$, yielding regret $O(K^{\frac{d+m+1}{d+2m+1}})$, which also approaches $O(\sqrt{K})$ as $m \to \infty$. With stronger assumption, our work also yields dimension-independent rates (see Corollary 3.6).
>
> Our bound is significant because it arises from pointwise guarantees on the per-step policy error, rather than only bounding the total cumulative regret. For a fixed horizon $H$, we show the expected policy error at each step $j$ across episode segments. Summing over steps yields the global regret in Eq. 21.
>
> Our estimates gives a much more fine-grained guarantees: rather than bounding only a global sum, we show that the learned policy is near-optimal at each timestep, avoiding issues like overfitting specific cumulative reward paths (e.g., reward hacking or physically inconsistent behavior). In this sense, pointwise bounds are stronger than bounding the total regret alone.
>
> **[Q1] Hyperparameter selection**
>
> For DPO, we use standard hyperparameters including a learning rate of 1e-3 and batch size of 32, without any special tuning. The algorithm performs well under these default settings and should not require extensive hyperparameter selection.
>
> We now provide two ablation tables examining sensitivity across key hyperparameters for benchmark algorithms.
>
> In the first table, we vary parameters:
> * CrossQ / TQC: number of critics and quantiles
> * SAC: entropy coefficient
>
> | Dataset         | DPO        | CrossQ        | CrossQ         | CrossQ         | SAC         | SAC               | SAC             | TQC         | TQC             | TQC             |
> |-----------------|------------|---------------|----------------|----------------|-------------|--------------------|------------------|-------------|------------------|------------------|
> | Ab. type   | original   | original      | n_critics=10   | n_critics=2    | original    | ent_coef=0.05      | ent_coef=0.2     | original    | n_critics=10     | n_quantiles=5    |
> | Surface   | **6.32**   | 6.42          | 7.33           | 6.63           | 7.41        | 7.62               | 8.23             | 6.67        | 6.68             | 6.96             |
> | Grid      | **6.06**   | 7.23          | 7.43           | 7.53           | 7.00        | 6.97               | 7.19             | 7.12        | 7.15             | 7.29             |
> | Mol Dyn   | **53.34**  | 923.90        | 1247.41        | 1287.99        | 1361.31    | 1367.50            | 1386.42          | 76.87       | 98.56            | 84.36            |
>
> In the second table, we vary:
> * DDPG: action noise type (Ornstein-Uhlenbeck), target update coefficient
> * PPO / TRPO: clip coefficient, Generalized Advantage Estimator parameter $\lambda$, and whether to normalize the advantage
>
> | Dataset         | DPO        | DDPG        | DDPG        | DDPG         | PPO         | PPO         | PPO                      | TRPO        | TRPO               |
> |-----------------|------------|-------------|-------------|--------------|-------------|-------------|---------------------------|-------------|---------------------|
> | Ablation type   | original   | original    | noise=OU    | tau=0.01     | original    | clip=0.1    | normalize_adv=False      | original    | GAE_lambda=0.8      |
> | Surface   | **6.32**   | 15.92       | 15.23       | 17.03        | 20.61       | 21.40       | 19.76                    | 6.48        | 11.67              |
> | Grid      | **6.06**   | 6.58        | 6.94        | 6.88         | 7.11        | 7.11        | 7.28                     | 7.10        | 7.19               |
> | Mol Dyn   | **53.34**  | 68.20       | 76.62       | 74.70        | 1842.31     | 1842.29     | 1842.31                  | 1842.28     | 1842.33            |
>
> Overall, we observe that varying hyperparameters across all algorithms does not substantially alter the performance ranking or lead to drastic performance shifts. Hence, we did not include these ablations in the original submission. Here we only show results for the reward-reshaped version here, but the straightforward (S-) version, omitted due to space, shows similar behavior and robustness across tasks.
>
> **[L1] Limitation on intuitive explanation**
>
> We will make revisions to wording and formatting to improve clarity and make the presentation more accessible to readers unfamiliar with control-theoretic concepts.
>
> We sincerely appreciate the reviewer’s thoughtful and constructive feedback. Due to the character limit, we have provided a concise response here, but we will carefully incorporate all suggestions and address the concerns raised in our final revision. Thank you again for your valuable insights and for engaging deeply with our work.
>
> **Reference**
>
> [Cai et al., 2020] Cai, Q., Yang, Z., Jin, C., Wang, Z.: Provably efficient exploration in policy optimization. In: Proceedings of the 37th International Conference on Machine Learning. Proceedings of Machine Learning Research, vol. 119, pp. 1283–1294. PMLR (13–18 Jul 2020)
>
> [Slivkins, 2022] Slivkins, A.: Introduction to multi-armed bandits (2022)
>
> [Vakili and Olkhovskaya 2024] Vakili, S., Olkhovskaya, J.: Kernelized reinforcement learning with order optimal regret bounds (2024)
>
> [Maran et al. 2024] Maran, D., Metelli, A.M., Papini, M., Restell, M.: No-regret reinforcement learning in smooth MDPs (2024)

---

> > ### Comment · Reviewer_X9Be · 2025-08-04
> >
> > I wish more justifications on tasks could be provided: e.g. why those tasks. I will maintain my score.

---

> > > ### Author Response · Authors · 2025-08-05
> > > **Justification for the selection of evaluation tasks**
> > >
> > > Thank you again for your time and thoughtful engagement with our work. Due to character limit in the official rebuttal, we were unable to fully elaborate on the motivations behind our task design. We hope that other concerns, including the addition of standard control tasks with reasonable performance from our algorithm, ablation studies, and the significance of our regret bounds, have been addressed in the rebuttal.
> > >
> > > In this comment, we focus on providing a more complete justification for the selection of evaluation tasks included in our submission.
> > >
> > > Our goal is to develop reinforcement learning methods with a focus on scientific computing applications. These problems often involve optimization over physical systems, where data simulation is expensive, and maintaining physical consistency during learning is crucial. Based on this motivation, we identified three representative and foundational task types:
> > >
> > > **Surface Modeling, control over geometries**
> > >
> > > At the level of an individual object, many scientific computing problems involve modifying the geometry of a structure to achieve desired physical properties. A standard example is the design of an airfoil (e.g., an aircraft wing), where the goal is to optimize its surface shape over time to minimize drag or maximize lift under aerodynamic flow. These surfaces are often altered through a set of control points, and the reward is derived from a functional measuring aerodynamic performance. Similarly, in structural engineering, surfaces can be automatically adjusted to improve stability against external disturbances, such as seismic vibrations. Additionally, in materials processing, surface optimization over time is used to control mechanical or thermal properties, such as stress tensor distributions or heat dissipation during the manufacturing of advanced materials. Our surface modeling task captures this family of problems by enabling control over geometries.
> > >
> > > **Grid-Based Modeling, control under PDE constraints**
> > >
> > > When moving beyond individual geometries to macro-scale physical systems, we typically encounter phenomena modeled by controlled partial differential equations (PDEs). These PDEs capture time-evolving quantities such as temperature, pressure, or concentration fields in space. For instance, the heat equation $\dfrac{\partial u(x, t)}{\partial t} = \Delta u(x, t) + f(x, t)$ on space variable $x$ and time variable $t$ models temperature evolution, where $u$ is the temperature field and $f$ is a control input. An important application is data center temperature control, where $f$ can represent electricity supplied to cooling elements, and the goal is to keep the temperature stable while optimizing the energy budget. Similar examples range from smart HVAC systems to industrial furnace regulation. Most, if not all, physical phenomena fall under this category and are represented by classical PDEs such as advection–diffusion equations, wave equations, reaction–diffusion systems, and elasticity equations. In one of our ongoing (non-theoretical) projects, we are working with the Black Oil equations, which govern fluid flow in petroleum reservoirs.
> > >
> > > In practical computational settings, solving such PDEs often requires spatial discretization, typically using a grid-based approximation. Due to computational constraints, control actions are applied on a coarser grid, while the underlying physical evaluation (i.e., computing the reward) is carried out on a finer grid that better captures the true dynamics. Our grid-based task precisely reflects this multiscale setting: it requires learning control policies that operate on a coarse discretization but are evaluated through a fine-grid reconstruction.
> > >
> > > **Molecular Dynamics**
> > >
> > > Now, at a much smaller atomic scale, such as those in molecular or biological systems, physical processes are often not well-described by a single PDE. Instead, one must work directly with the atomic structures, whose interactions are governed by complex, often nonlocal, energy-based potentials. This motivates our third category of molecular dynamics. One example is understanding how virus capsids optimally change over time under therapeutic molecular interactions. This is important for designing more effective treatments.
> > >
> > > In summary, our three evaluation tasks correspond to core abstractions in scientific computing. As summarized in the main paper and reiterated during the rebuttal, these include:
> > >  * Optimization over geometric surfaces,
> > >  * Grid-based modeling with controlled PDEs, and
> > >  * Molecular dynamics in atomistic systems.
> > >
> > > We hope this more detailed explanation partly addresses your concern and would be happy to engage further if anything remains unclear. We sincerely appreciate your feedback, which has helped clarify and strengthen the presentation and impact of our work.

---

### Note · Authors · 2025-08-13

We thank all reviewers for their thoughtful reviews, valuable suggestions, and for taking time to read our paper. We particularly appreciate the positive recognition of many aspects of our work, including novelty/significance (X9Be, dPSd, HKCT), clarity (TLm5, dPSd), strong theoretical foundation (dPSd, HKCT), and sound proofs (dPSd, HKCT).

We propose a novel learning framework using continuous-time control formulation and its Hamiltonian differential dual. We then develop a policy optimization algorithm that provides fine-grained control updates aligned with system dynamics. Our theoretically-grounded approach with rigorous pointwise convergence and competitive regret is particularly well-suited for RL problems requiring sample efficiency and physical consistency, such as those in scientific computing domains.

As scientific computing spans a diverse range of problems, abstracting them into 3 classes enables a unified mathematical treatment, with our three tasks as representative instances: (i) individual objects (surface control), (ii) macro-scale systems governed by PDEs (grid-based modeling), and (iii) atomistic systems (molecular dynamics). Experiments on those show consistent gains over widely used RL methods, despite extra baseline tuning.

Based on the reviewers' feedback, we provided detailed rebuttal responses and additional comments to help clarify/strengthen our contributions, including:

**(1) Related work/positioning**. We contextualize our unique approach within continuous-time RL. Our regret bound is notable as it arises from pointwise guarantees on per-step policy error, rather than bounding only global sum, thereby avoiding overfitting to specific reward paths that cause inconsistent behaviors.

**(2) Ablations**. Ablations on key hyperparameters for strong baselines show no net gains, highlighting our method’s effectiveness.

**(3) Standard tasks**. We report reasonable performance on standard RL tasks, in addition to our strong performance in scientific computing, where learning challenges remain for standard benchmarks.

**(4) Compute footprint**. Time complexity, memory, training times are in Sec. 4.2, App. C.2, and rebuttal, respectively.

We will fix editing issues, add ablation study and related work as outlined during rebuttal to the final version, plus further computational experiments (as allowed) in the appendix.

We sincerely appreciate your discussions and feedbacks, which help refine this work and guide future directions.

---

### Decision · Program_Chairs · 2025-09-17

**Decision:**

Accept (poster)

**Comment:**

This paper presents DPO, a novel framework that reformulates reinforcement learning control through differential and pointwise constraints, offering both theoretical guarantees and practical benefits. The reviewers initially raised concerns about clarity and the strength of empirical validation, but the authors’ rebuttal and discussion addressed these points effectively by clarifying the motivation, refining the exposition, and providing additional experimental details. Overall, reviewers agreed that the work introduces a fresh and principled perspective on RL control, with rigorous analysis and encouraging empirical results that distinguish it from incremental extensions. The consensus after discussion was positive: while broader experiments would further strengthen the paper, the contribution is seen as technically solid, conceptually novel, and relevant to the community. I therefore recommend acceptance.